# BINDER: HIERARCHICAL CONCEPT REPRESENTATION THROUGH ORDER EMBEDDING OF BINARY VECTORS

## ABSTRACT

For natural language understanding and generation, embedding concepts using an order-based representation is an essential task. Unlike traditional point vector based representation, an order-based representation imposes geometric constraints on the representation vectors for explicitly capturing various semantic relationships that may exist between a pair of concepts. In existing literature, several approaches on order-based embedding have been proposed, mostly focusing on capturing hierarchical relationships; examples include Order Embedding, Poincaré embedding on hyperbolic space, and Box Embedding. Box embedding creates region-based rich representation of concepts, but along the process it sacrifices simplicity, requiring a custom-made optimization scheme for learning the representation. Poincaré embedding improves embedding quality by exploiting the ever-expanding property of hyperbolic space, but it also suffers from the same fate as box embedding as gradient descent like optimization is not simple in the hyperbolic space. In this work, we propose BINDER, a novel approach for order-based representation. BINDER uses binary vectors for embedding, so the embedding vectors are compact with order of magnitude smaller footprint than other methods. BINDER uses a simple and efficient optimization scheme for learning representation vectors with a linear time complexity. Our comprehensive experimental results show that BINDER is very accurate, yielding perfect accuracy on the reconstruction task. Also, BINDER can learn concept embeddings just from the direct edges, whereas all existing order-based approaches rely on indirect edges. For example, BINDER achieves **18%** higher accuracy (98.4%) over the second best Order Embedding model for our WordNet Nouns dataset (with 743,241 edges) in 0% transitive closure experiment setup.

## 1 INTRODUCTION

Ontologies transcribe the knowledge of a domain through formal listing of concepts along with various semantic relations that may exist among the concepts. The most important among these relations is hypernym-hyponym, which captures the *is-a* relation between a specialized and a general concept of a domain. Such knowledge are essential for achieving optimal results in various natural language generation tasks, such as image caption generation (Karpathy & Fei-Fei, 2015; Vinyals et al., 2015), question-answering (Dai et al., 2016; Ren et al., 2020; Yao et al., 2019), and taxonomy generation (Nakashole et al., 2012; Wu et al., 2012). For example, using an appropriate ontology, an image caption generator can opt for a generic text "a person is walking a dog" instead of a more informative text, "a woman is walking her dog", if the model is not very confident about the gender of the person walking the dog. However, building a large conceptual ontology for a domain is a difficult task requiring gigantic human effort, so sophisticated machine learning models, which can predict *is-a* links between pairs of concepts in an ontology, is of high demand.

The task of predicting is-a relationship between two concepts in an ontology can be viewed as a link prediction task in a concept graph, or formally speaking, in an ontology chart. Although link prediction (Hasan & Zaki, 2011) is a well-studied task, it did not receive much attention by the NLP or machine learning researchers. A good number of works (Galárraga et al., 2013; Lao et al., 2011) address link prediction in knowledge graphs, but they are customized for the Resource Description Framework (RDF) type of data. Lately, node embedding using shallow (Perozzi et al., 2014; Tang et al., 2015; Grover & Leskovec, 2016), or deep neural networks (Guo et al., 2019; Neelakantan

et al., 2015; Schlichtkrull et al., 2017; Shang et al., 2018; Nguyen et al., 2018; Yao et al., 2019) have shown improved performance for solving link prediction tasks. The majority of these works consider undirected graphs, so they are not suitable for predicting edges in a concept graph. Many works (Bordes et al., 2013; He et al., 2015; Lin et al., 2015; Sun et al., 2019; Trouillon et al., 2016; Wang et al., 2014) considered node embedding for knowledge graphs, where embedding of head nodes, tail nodes, and relation nodes are learned, but such works are only suitable for RDF triple based data representation.

In recent years, some progress has been made along embedding concepts which may have some semantic relationships. One of the earliest efforts in this direction is a work by Vendrov et al. (2015), which proposed order embedding. The main idea is to embed the concepts in the positive cone $\mathbb{R}_+^d$. In this embedding, if $a$ is-a $b$, then their corresponding embedding vectors satisfy $f(b) \leq f(a)$ element-wise. In this way, along all dimensions, the value of $f(b)$ is smaller than that of $f(a)$, and a generic concept hovers more closer to the origin with smaller norm than the associated specialized concept. In another line of works (Nickel & Kiela, 2017; Ganea et al., 2018), which focus on embedding trees, DAGs or tree-like graphs, hyperbolic space is used instead of Euclidean space. In hyperbolic space, two non-intersecting straight lines diverge, allowing for the embedding of more objects along the periphery. It is beneficial for embedding a tree structure which has exponentially more nodes at a higher depth. The third line of work, known as box embedding (Vilnis et al., 2018; Li et al., 2019; Dasgupta et al., 2020; 2021; Boratko et al., 2021), deviates from vector (point) embedding, and instead uses a rectangular region for embedding a concept. This representation is richer as it both helps embedding order relation and part-of relations between concepts, or overlapping concepts, thereby overcomes some of the limitations of earlier approaches.

In this work, we propose an order embedding model that is simple, elegant and compact, and does not suffer from the above limitations that we discussed about the existing order embedding schemes. Our idea is to use binary vectors for order embedding, i.e. for each entity $a$, $f(a) = \mathbf{a} \in \{0,1\}^d$. In other words, we embed each object at a vertex of a $d$-dimensional non-negative unit hypercube, where $d$ is a user-defined parameter. The overall approach is simple, as for denoting $a$ is-a $b$, we only require that $f(b)_j = 1 \implies f(a)_j = 1, \forall j \in [1:d]$, i.e., along any embedding dimension $j$, if $\mathbf{b}_j$ is 1, then $\mathbf{a}_j$ must be 1. The idea is fairly intuitive; if we consider each dimension denoting some latent property which make something $b$, given that $a$ is-a $b$, $a$ also have those properties. Since it uses bits for embedding, the embedding vectors are compact with order-of-magnitude smaller memory footprint (see Appx. F.5) compared to other methods. Our embedding idea is elegant as it captures the is-a relation through intent-extent philosophy of formal concept analysis (FCA) (Ganter & Wille, 1999), which is a principal way of deriving a concept hierarchy.

The major challenge for our proposed embedding idea is finding an effective optimization algorithm for learning the embedding, as we deviated away from continuous Euclidean space and embraced a combinatorial space. In this sense, given the training data (a collection of hyponym-hypernym pairs), learning the embedding of the objects in the training data becomes a classical combinatorial feasibility task, a known NP-complete problem. We use a randomized local search algorithm inspired by stochastic gradient descent for solving this problem. Our optimization uses a "gradient" for each bit position to calculate a probability of flipping that bit. The idea of using gradient to flip bit probabilistically to optimize in binary space is certainly novel. The novelty of our optimization method from methodology perspective consists of computing a proxy of gradient for a binary space (Section 2.4) and flip probability (Section 2.5). The contribution of this work is innovative, and novel, both in terms of embedding idea and optimization framework. Our algorithm is very fast; the overall computational complexity is $O(ndT(|P| + |N|))$, for $n$ words and $d$ dimensions, which is linear in each variable. We name our method BINDER [1]. We claim the following contributions:

1. We propose BINDER, a novel order embedding approach which embeds the entities at the vertex of a $d$-dimensional hypercube. We show that BINDER is ideal for finding representation of entities or concepts which exhibit hyponym-hypernym relationship. BINDER is simple, compact, efficient, and has better generalization capacity over transitive edges compared to existing methods in a transductive setting.

---

[1]The name Binder is an abbreviation of **Bin**ary Or**der** Embedding.

2. BINDER uses a novel local search based optimization algorithm for solving the embedding learning task. The proposed algorithm is simple, efficient and effective, and a proxy of gradient descent for the combinatorial space.

3. Experiments on five benchmark datasets show that BINDER exhibits superior performance than the existing state-of-the-art algorithms on transitive closure link prediction and reconstruction tasks.

## 2 BINARY ORDER EMBEDDING (BINDER)

### 2.1 MOTIVATION

Our main motivation is to embed entities that have hierarchical (is-a) relations between pairs of them. For this, one must impose order between two embedding vectors. For representing $x$ is-a $y$, one earlier work (Vendrov et al., 2015) has imposed order by requiring $x_i \leq y_i, \forall i \in [1 : d]$ for embedding vectors $x$ and $y$ in real space. BINDER uses a similar idea, but instead it uses binary space, in which $x_i \leq y_i$ becomes $x_i \implies y_i$. Implication obeys the transitive property, so BINDER's binary representation works well for is-a relationship, which is transitive. BINDER has the following benefits: (1) Binary representations are compact, often taking order of magnitude less memory than real space embedding, and is computationally efficient (demonstrated in Appendix F.5). (2) Binary representation can immediately provide representation of concepts that can be obtained by logical operation over the given concept vectors. For instance, given vectors for the concepts "vehicle" and "flying", we can find a subtree of flying vehicles by taking the union of the *vehicle* and *flying* vectors. Or, if we have representation vectors for "men's shoe" and "women's shoe", and we want a representation vector for shoe, we can obtain that by taking the intersection of the above two vectors. Such operation can be extended to any complex Boolean operations. (3) Binary representation is explainable in the sense that we can treat the binary representation as a set of latent properties; a "1" value at a dimension means that the entity possesses that property, and a "0" value means that it does not possess that property. Using this argument we can obtain explainable embeddings, which is not immediately available for vectors in real space. We give a small demonstration of this in Figure 1 in appendix D, where we trained our model on a small lattice. In particular, the embedding, being a binary matrix, can be thought of as a machine-learned object-attribute table. Such a table can be further studied by using formal concept analysis (FCA). BINDER's representation not only provides a simple check of two concept's subsumption relation, it also provides a intuitive distance measure between two related concepts. If $a$ is-a $b$, the Hamming distance of their representation gives us an indication of their distance. The number of '1's in a concept's representation provides an indication of the narrowness of a concept. Using domain knowledge, and by observing the distribution of '1's in a column, one can further deduce which dimension may represent which of the properties (intent). Above all of these, our experiments show that binary representation vectors also perform well on reconstruction and link prediction tasks.

### 2.2 PROBLEM FORMULATION

For a concept $a$, BINDER embeds $a$ through a $d$-dimensional binary vector, so every concept is embedded at the vertex of a $d$-dimensional non-negative unit hypercube, where $d$ is a user-defined parameter. If $a$ is-a $b$, and in the embedding space $\mathbf{a}$ and $\mathbf{b}$ are their representation, then BINDER satisfies $\mathbf{b}_k = 1 \implies \mathbf{a}_k = 1, \forall k \in \{1, \ldots, d\}$. This embedding idea entails from the observation that when $a$ is-a $b$, $a$ must possess all the properties that $b$ possesses. In BINDER's embedding, an '1' in some representation dimension denotes "having a latent property"; say $j$'th dimension of $\mathbf{b}$ has a value of 1, i.e., $b$ possesses the property $j$, then $a$ must also possess the property $j$, which is captured by the above requirement.

To learn embedding vectors for a collection of concepts in a domain, BINDER uses a supervised learning approach. Given a set of concepts $W$ and partial order concept-pairs $P = \{(a, b) : a \text{ is-a } b\}$, BINDER's task is to find an embedding function $B : W \rightarrow \{0, 1\}^d$ such that for any $a, b \in W$,

$$(\mathbf{a} \cap \mathbf{b}) = \mathbf{b} \text{ iff } (a, b) \in P \text{ and } a \neq b \tag{1}$$

holds; here $\mathbf{a} = B(a)$ and $\mathbf{b} = B(b)$ are the embedding vectors for concepts $a$ and $b$, and $\cap$ denotes the bitwise AND operation.

The above learning task is a constraint satisfaction problem (CSP) or a feasibility task in the combinatorial space, which is a known NP-Complete task (Cormen et al., 2022). To solve it, BINDER uses a randomized local search algorithm, which is fast and effective. Note that given a training dataset, $P$, if BINDER's embedding solution satisfies all the constraints, then the embedding is perfect and all the partial order pairs in $P$ can be reconstructed from the embedding with 100% accuracy. But the goal of our embedding is not necessarily yielding a 100% reconstruction accuracy on the training data, rather to perform is-a prediction task on an unseen test dataset, so we do not strive to solve the CSP task exactly. In the next section, we discuss BINDER's learning algorithm.

**Notations:** Italic letters such as $a, b$ denote entities, while boldface $\mathbf{a}, \mathbf{b}$ denote their embedding vectors. $\mathbf{a}_j$ denotes the value at the $j$'th position in $\mathbf{a}$. In the algorithms, we use $B$ to denote the complete binary embedding matrix, $B[a, :]$ for $\mathbf{a}$, and $B[a, j]$ for bit $j$ of said vector. We use $*$ to denote element-wise multiplication; all arithmetic operations are done in $\mathbb{Z}$ or $\mathbb{R}$. Finally, we write pairs in hyponym-hypernym order: a pair $(a, b)$ refers to the statement "$a$ is-a $b$".

## 2.3 TRAINING ALGORITHM

The learning task of BINDER is a CSP task, which assigns $|W|$ distinct binary $d$-bit vectors to each of the variables in $W$, such that each of the constraints in the partial order $P$ is satisfied. Various search strategies have been proposed for solving CSP problems, among which local search and simulated annealing are used widely (Beck et al., 2011). For guiding the local search, we model this search problem as an optimization problem, by designing a loss function that measures the fitness of an embedding. A simple measurement is the number of pairs in $P$ that violates the constraint in Equation 1. Note that the constraint is "if and only if", which means for any pair $(a', b')$ that is not in $P$, we want $\mathbf{a}'$ and $\mathbf{b}'$ to *not* satisfy this constraint. If $|W| = n$, we have exactly $|P|$ constraints for the positive pairs (we call these *positive constraints*), and $n^2 - n - |P|$ *negative constraints* for the negative pairs. Using these constraints, we compute a simple loss function—a linear function of the number of violated positive and negative constraints as shown below:

$$Loss_P = \alpha \sum_{(a,b) \in P} \sum_j \mathbf{1}_{(\mathbf{a}_j, \mathbf{b}_j) = (0,1)}(a, b) \tag{2}$$

$$Loss_N = \beta \sum_{(a',b') \in N} \mathbf{1}_{\forall j (\mathbf{a}'_j, \mathbf{b}'_j) \in \{(0,0),(1,0),(1,1)\}}(a', b') \tag{3}$$

$$Loss = Loss_P + Loss_N, \tag{4}$$

where $\alpha$ and $\beta$ are user-defined parameters and $\mathbf{1}$ is the indicator function. Due to the above loss function, BINDER's learning algorithm relies on the existence of negative pairs $N \subseteq \{(a', b') : a' \text{ is-not-a } b'\}$. If these negative pairs are not provided, we generate them by randomly corrupting the positive pairs $P$ as in Vendrov et al. (2015); Nickel & Kiela (2017); Ganea et al. (2018), by replacing $(a, b) \in P$ with $(r, b)$ or $(a, r)$ where $r$ is sampled randomly from the entity set $W$.

For local search in *continuous* machine learning, the search space is explored using some variant of gradient descent. This gradient is defined as the derivative of the loss function (or an approximation thereof) with respect to each parameter. With binary vectors, the parameters are discrete, but we can get a proxy of the "gradient" by taking the *finite difference* between the current value of the loss function, and the new value after a move is made. In a continuous space, vectors can be updated by adding or subtracting a delta, but in discrete space, the new vector is one of the neighboring vectors where the neighborhood is defined explicitly. If the neighborhood is defined by unit Hamming distance, we can make a neighbor by flipping one bit of one vector, but for large datasets, such an approach will converge very slowly. In BINDER, we randomly select bits to be flipped by computing a probability from the gradient value of each bit position, as shown in the following subsection.

## 2.4 GRADIENT DERIVATION

BINDER's gradient descent scheme is based on correcting order of positive pairs by flipping bits, which are chosen randomly with a probability computed from a discrete gradient value. Below we discuss how this gradient is computed.

A sub-concept will share all the attributes (bits set to 1) of the concept, and possibly contain more attributes. For each positive pair $(a, b) \in P$ and each bit index $j$, we aim to avoid having $(\mathbf{a}_j, \mathbf{b}_j) =$

Table 1: Logic truth table to flip bits in positive (first 3 columns) and negative (last 3 columns) pairs

| $\mathbf{a}_j$ | $\mathbf{b}_j$ | $a$ is-a $b$ | $\mathbf{a}'_j$ | $\mathbf{b}'_j$ | $a'$ is-not-a $b'$ |
|---|---|---|---|---|---|
| 0 | 0 | Protect $\mathbf{b}_j$ | 0 | 0 | Flip $\mathbf{b}'_j$ |
| 0 | 1 | Flip either bit | 0 | 1 | Protect both bits |
| 1 | 0 | Don't care | 1 | 0 | Flip both $\mathbf{a}'_j$ and $\mathbf{b}'_j$ |
| 1 | 1 | Protect $\mathbf{a}_j$ | 1 | 1 | Flip $\mathbf{a}'_j$ |

Table 2: Logic truth table to calculate positive and negative loss gradient

| $\mathbf{a}_j$ | $\mathbf{b}_j$ | $\Delta_{\mathbf{a}_j Loss_P}$ | $\Delta_{\mathbf{b}_j Loss_P}$ | Comments | $\mathbf{a}'_j$ | $\mathbf{b}'_j$ | $\Delta_{\mathbf{a}'_j Loss_N}$ | $\Delta_{\mathbf{b}'_j Loss_N}$ | Comments |
|---|---|---|---|---|---|---|---|---|---|
| 0 | 0 | 0 | $-1$ | Protect $\mathbf{b}_j$. | 0 | 0 | 0 | 1 | Flip $\mathbf{b}'_j$. |
| 0 | 1 | 1 | 1 | Flip $\mathbf{a}_j$ / $\mathbf{b}_j$* | 0 | 1 | 0 | 0 | Don't care |
| 1 | 0 | 0 | 0 | Don't care | 1 | 0 | 0 | 0 | Don't care |
| 1 | 1 | $-1$ | 0 | Protect $\mathbf{a}_j$. | 1 | 1 | 1 | 0 | Flip $\mathbf{a}'_j$. |

\* or both bits

$(0, 1)$, since that would imply $a$ did not inherit attribute $j$ from $b$. On the other hand, for negative pairs $(a', b')$, we aim to create at least one bit index with $(0, 1)$ bit pair. Suggested bit flipping or protecting operations for these requirements are shown in the third and sixth column of Table 1; for $a$ is-a $b$, we do not want $(0, 1)$ configuration, hence we protect $\mathbf{b}_j$ in first row, $\mathbf{a}_j$ in the forth row, and flip either bit in the second row. On the other hand, for $a$ not-is-a $b$, we want a $(0, 1)$ configuration. If the model currently suggests $a'$ is-a $b'$ (i.e. there is no $j$ where $\mathbf{a}'_j = 0, \mathbf{b}'_j = 1$), we correct this by flipping either the $\mathbf{a}'$ side of a $(1, 1)$ pair (fourth row) or the $\mathbf{b}'$ side of a $(0, 0)$ pair (first row), as shown in the sixth columns of the same table. Note that negative samples are needed so we can avoid trivial embeddings, such as all words being assigned the zero vector.

We first derive the gradient of $Loss_P$ with respect to $\mathbf{a}_j$ and $\mathbf{b}_j$. We define the "gradient" $\Delta_{\mathbf{a}_j}$ to be positive if flipping bit $\mathbf{a}_j$ *improves* the solution according to our loss function, regardless of whether bit $\mathbf{a}_j$ is currently 0 or 1. As shown in Column 3 of Table 2, flipping $\mathbf{a}_j$ makes no change in loss for the first and third rows; but for the second row, one violation is removed, and for the fourth row, one new violation is added. For the four binary bit configurations, $\{(0, 0), (0, 1), (1, 0), (1, 1)\}$, the improvement to $Loss_P$ is 0, 1, 0, and -1, respectively, as shown in Column 3 of Table 2. Column 4 of the same table shows the value of $\Delta_{\mathbf{b}_j} Loss_P$ calculated similarly. It is easy to see that these two sets of gradients values can be written as $\mathbf{b}_j(1 - 2\mathbf{a}_j)$ (for values in Column 3) and $(1 - \mathbf{a}_j)(2\mathbf{b}_j - 1)$ (for values in Column 4), respectively. Now we can use the following equations which shows the gradient expression for all the dimensions of the bit vector, for all positive pairs, $(a, b) \in P$.

$$\Delta_{\mathbf{a}} Loss_P = \alpha \sum_{b: (a,b) \in P} \mathbf{b} * (1 - 2\mathbf{a}) \quad (5) \qquad \Delta_{\mathbf{b}} Loss_P = \alpha \sum_{a: (a,b) \in P} (1 - \mathbf{a}) * (2\mathbf{b} - 1) \quad (6)$$

$\Delta_{\mathbf{a}} Loss_P$ and $\Delta_{\mathbf{b}} Loss_P$ are $d$-dimensional integer vector, where $d$ is the embedding dimension. Now, to calculate negative loss gradient, we count the number of $(\mathbf{a}'_j, \mathbf{b}'_j) = (0, 1)$ "good" bit pairs in a negative pair. If $G_{a',b'} := \sum_j \mathbf{1}_{(\mathbf{a}'_j, \mathbf{b}'_j) = (0,1)}(a', b') = 0$, then $(a', b')$ is a false positive, so we need to flip a bit to make $(\mathbf{a}'_j, \mathbf{b}'_j) = (0, 1)$. So for the case of violation, based on the bit values in Columns 8 and 9 of Table 2, we derived the following algebraic expressions:

$$\Delta_{\mathbf{a}'} Loss_N = \beta \sum_{\substack{b': (a',b') \in N, \\ G=0}} \mathbf{a}' * \mathbf{b}' \quad (7) \qquad \Delta_{\mathbf{b}'} Loss_N = \beta \sum_{\substack{a': (a',b') \in N, \\ G=0}} (1 - \mathbf{a}') * (1 - \mathbf{b}') \quad (8)$$

On the other hand, if $G_{a',b'} = 1$, there is no violation, but the not-is-a relation is enforced by only one bit, so that bit must be protected. That is, if $(\mathbf{a}'_j, \mathbf{b}'_j) = (0, 1)$ for exactly one $j$, then we want the gradient to be $-1$ for that index $j$ (recall that negative gradient means flipping is bad), and zero gradient for all other indices. This is true for exactly the bit $j$ satisfying $b'_j(1 - a'_j) = 1$, so we have

$$\Delta_{\mathbf{a}'} Loss_N = -\beta \sum_{\substack{b': (a',b') \in N, \\ G=1}} \mathbf{b}' * (1 - \mathbf{a}') \quad (9) \qquad \Delta_{\mathbf{b}'} Loss_N = -\beta \sum_{\substack{a': (a',b') \in N, \\ G=1}} \mathbf{b}' * (1 - \mathbf{a}') \quad (10)$$

For $G_{a',b'} > 1$, the gradient is 0 for all bit index, as no bits need immediate protection. Finally, we linearly add gradients over $Loss_P$ and $Loss_N$ to get the final gradient matrix, $\Delta = \Delta^+ + \Delta^-$. The overall process is summarized in Algorithm 1 in Appendix B. In this Algorithm, $\Delta^+$ is the gradient of $Loss_P$ and $\Delta^-$ is the gradient of $Loss_N$. $\Delta$, $\Delta^+$ and $\Delta^-$ all are integer-valued matrices of size $n \times d$, where $n$ is the vocabulary size and $d$ is the embedding dimension.

## 2.5 Flip probability

In binary embedding, each bit position takes only two values, 0 and 1. Traditional gradient descent, which updates a variable by moving towards the opposite of gradient, does not apply here. Instead, we utilize randomness in the update step: bit $j$ of word $w$ is flipped with a probability based on its gradient, $\Delta[w, j]$. To calculate the flip probability we used $\tanh$ function. For each word $w$ and each bit $j$ we compute the gradient $\Delta[w, j]$ as in Algorithm 1 and output

$$\text{FlipProb}[w, j] = \max\left\{0, \tfrac{1}{2}\tanh\left(2(r_\ell\,\Delta[w, j] + b_\ell)\right)\right\} \tag{11}$$

where the learning rate $r_\ell$ is used to control frequency of bit flips, and the learning bias $b_\ell$ makes sure bit vector update does not get stuck if all probability values are 0. The division by 2 prevents the model from flipping (on average) more than half of the bits in any iteration; without it, the model would sometimes flip nearly every bit of a vector, which will cause the model to oscillate. Therefore, we (softly) bound probabilities by $\frac{1}{2}$ to maximize the covariance of the flips of each pair of vectors. The inside multiplication by 2 preserves the property $\frac{1}{2}\tanh(2p) \approx p$ for small $p$, making hyperparameter selection more intuitive: $b_\ell$ is (almost exactly) the probability of flipping a neutral bit, and $\alpha r_\ell$ approximates the increase in probability for each positive sample that would be improved by flipping a given bit (and likewise for $\beta r_\ell$ and negative samples). We note that the three hyper parameters $r_\ell$, $\alpha$, and $\beta$ are somewhat redundant, since we can remove $r_\ell$ and replace $\alpha$ and $\beta$ with $r_\ell\alpha$ and $r_\ell\beta$ respectively without changing the probability. The only reason for using three parameters is to keep the $\alpha$ and $\beta$ computations in integer arithmetic for faster speed.

## 3 Experiments and Results

To demonstrate the effectiveness of BINDER, we evaluate its performance on Reconstruction and Link Prediction tasks, which are popular for hierarchical representation learning. We use the same experimental setup as the existing methodologies and compare BINDER's performance with them.

**Reconstruction** is an inverse mapping task from embedding vectors to the list of positive and negative pairs in the training data. A high accuracy in reconstruction testifies for the capacity of the learning algorithm: it confirms that BINDER's learning algorithm can obtain embedding vectors to satisfy the representation constraints of both the positive and negative pairs, as defined in the problem formulation (Section 2.2). To evaluate representation capacity, we train BINDER over the full transitive data set. We then create a test dataset which includes the entire positive edge set in the training data, and a random set of negative samples of the same size, ensuring that these negative samples are unique and not in the positive set (this setup is identical to Vendrov et al. (2015)). We then validate whether, for each positive pair $(a, b)$, their bit embeddings $\mathbf{a}$ and $\mathbf{b}$ satisfy $(\mathbf{a} \cap \mathbf{b}) = \mathbf{b}$ and $\mathbf{a} \neq \mathbf{b}$. If so, the corresponding entry is correctly reconstructed. Then, for each negative pair $(c, d)$ we confirm whether $(\mathbf{c} \cap \mathbf{d}) \neq \mathbf{d}$ is true. If yes, this pair is correctly reconstructed.

**Link Prediction** task is predicting edges that the learning algorithm has not seen during training. For this task, we split the dataset into train, test and validation. We randomly select a portion of edges for the test split, and an equal number for the validation set, by randomly holding out observed links. Details of the datasets can be found in Section 3.1. Since is-a relation is transitive, in theory, transitive-pair edges are redundant: if $(a, b)$ and $(b, c)$ are positive pairs, then $(a, c)$ should also be a positive pair. A good embedding model should be able to deduce such pairs automatically.

The remainder of this Section is as follows. In Section 3.1, we explain the datasets. In Section 3.2, we discuss competing methods and our justification of why they were chosen. Then we compare BINDER's performance with those of the competing methods, and give our results in Sections 3.3 and 3.4). Further details are given in the appendices. In Appendix C, we discuss low-level details of training and hyperparameter tuning. In Appendix D, we show a visual representation of BINDER's embedding in a small dataset. We also perform an ablation study to show how different component of cost function affects BINDER's performance; results of ablation study is provided in Appendix F.1.

### 3.1 Datasets

For reconstruction task, we evaluate our model on **5** datasets. We downloaded Music and Medical domain dataset from SemEval-2018 Task 9: Hypernym Discovery. We collected Lex and random

Table 3: Reconstruction Results   Balanced Acc(%) (dim)

| Model | Medical
entities = 1.4k
edges = 4.3k | Music
entities = 1k
edges = 6.5k | Shwartz
Lex
entities = 5.8k
edges = 13.5k | Shwartz
Random
entities = 13.2k
edges = 56.2k | WordNet
Nouns
entities = 82k
edges = 743k |
|---|---|---|---|---|---|
| OE | **100** (10) | **100** (20) | 100 (20) | 100 (20) | 97.4 (200) |
| Poincaré | 92.6 (10) | 88.3 (20) | 95.3 (20) | 92 (100) | 97.2 (50) |
| HEC | 95.7 (20) | 92.1 (100) | 99 (5) | 96.4 (100) | 91.1 (100) |
| T-Box | 100 (25) | 100 (50) | 100 (25) | 100 (50) | 99.8 (50) |
| BINDER | 99.9 (50) | 100 (100)* | **100** (50)* | **100** (50)* | **99.9** (80) |

*For Shwartz Lex and Random dataset, BINDER dimension is higher compared to OE but considering space complexity (1 bit vs 4 bytes) for each dimension we conclude BINDER as the wining model.

dataset from Shwartz et al. (2016) which were constructed by extracting hypernymy relations from WordNet (Fellbaum, 2012), DBPedia (Auer et al., 2007), Wiki-data (Vrandecic, 2012) and Yago (Suchanek et al., 2007). The largest dataset is the WordNet noun hierarchy dataset. The full transitive closure of the WordNet noun hierarchy consists of 82,115 Nouns and 743,241 hypernymy relations. This number includes the reflexive relations $w$ is-a $w$ for each word $w$, which we removed for our experiments, leaving 661,127 relational edges. We generate *negative samples* following a similar method to Order Embedding (Vendrov et al., 2015) paper: we corrupt one of the words in a positive pair, and discard any corrupted pairs that happen to be positive pairs.

For prediction task, we evaluate our model on the same datasets. Our dataset creation and experimental setup are identical to Hyperbolic Entailment Cones paper (Ganea et al., 2018). We always include the direct edges, edges $(a, b)$ with no other word $c$ between them, in the training set. The remaining "non-basic" edges (e.g. 578,477, for WordNet Nouns dataset) are split into validation (5%) and test (5%). We generate four training sets that include 0%, 10%, 25% and 50% of the remaining non-basic edges which are randomly selected. For validation and test set, we use 10 times more negative pairs than positive pairs, so to keep consistency we re-weight the negative data by $1/10$ before evaluating accuracy. We remove the root of WordNet Nouns dataset as it has trivial edges to predict. The remaining transitive closure of the WordNet Nouns hierarchy consists of 82,114 Nouns and 661,127 hypernymy relations.

## 3.2   COMPETING METHODS AND METRICS USED

We compare our model to four existing order embedding methods: the continuous Order Embedding (Vendrov et al., 2015), Poincaré embeddings (Nickel & Kiela, 2017), Hyperbolic Entailment Cones (Ganea et al., 2018), and the T-Box (Boratko et al., 2021). All four methods are intended to produce embedding for entities having hierarchical organization. Among the above methods, our model is most similar to Order Embedding Vendrov et al. (2015), as our model is simply the restriction of theirs from $(\mathbb{R}^+)^d$ to the set $\{0, 1\}^d$. So, this model is a natural competitor. Similarity of our model to Hyperbolic Entailment Cones Ganea et al. (2018) is that both are transitive order embeddings. We also compare to Poincaré embeddings Nickel & Kiela (2017), which makes use of a distance ranking function to embed the undirected version of the hierarchy graph. Finally we compare with the latest region-based probabilistic box embeddings with temperatures Boratko et al. (2021). In subsequent discussion and result tables, we will refer to order embedding as OE, Poincaré embedding as Poincaré, hyperbolic entailment cones as HEC and probabilistic box embeddings with temperatures as T-Box. We report *balanced* accuracy, meaning we either ensure the positive and negative test sizes are equal (for reconstruction) or re-weight the negative samples (for transitive prediction). We also report F1-score of all experiments in Appx. F.2.

BINDER, being discrete, has the additional property that two distinct words have a non-trivial chance of being assigned the same embedding. For this reason, we evaluate BINDER on the *strict* hypernym relation: we predict an edge $(a, b)$ as positive (hypernym) if and only if $(\mathbf{a} \cap \mathbf{b}) = \mathbf{b}$ (as in Eq. 1) *and* $\mathbf{a} \neq \mathbf{b}$. This is irrelevant for competing models, for which exact equality is extremely unlikely.

### 3.3 Reconstruction Task Results

This task is relatively easy; BINDER and all the competing methods perform better in this task than in the link prediction task. This is because reconstruction is similar to training accuracy and link prediction task is similar to test accuracy in a typical machine learning task. The results are shown in Table 3, in which the datasets are arranged in increasing size from left to right column-wise. For each dataset and each model, we show accuracy and the dimension (in parenthesis) for which that accuracy was achieved. As we can see, BINDER achieves near perfect result for each dataset and maintains its consistent performance for larger datasets. Among the competitors, T-Box performs the best with 100% accuracy for all datasets except the largest Nouns dataset (99.8%). OE performs the third best; it has a perfect 100% accuracy for relatively smaller datasets, but its performance drops by 3 percent for the larger dataset i.e. WordNet Nouns. HEC performs slightly better for smaller datasets than Poincaré. Compared to other tasks, we found that reconstruction performance generally improves as we increase the dimension. It makes sense, as reconstruction is like training accuracy, and with higher dimensions, models have more freedom (higher capacity) to satisfy the order constraints listed in the training data. Note that BINDER is a randomized algorithm, so we show the best results that we obtain in 5 runs. The mean and standard deviation of its performance metrics is shown in Table 5.

### 3.4 Link Prediction Results

Link prediction is more critical than Reconstruction, as it is made on unseen data. We execute multiple variants of this task with different degree of transitive edges (0%, 10%, 25%, and 50%) in the training data. Obviously, 0% transitive edge scenario is the most difficult. The results are shown in Table 4. In this table also, we show accuracy along with the dimension value for which the best accuracy is obtained. As we can see from this table, BINDER handsomely wins over all competitors in the 0% transitive closure link prediction task. As the dataset size increases, the accuracy margin over the second best also increases. For the largest dataset, WordNet Nouns, BINDER achieves 18% better accuracy and 42% better $F_1$ score than the second best. Note that in a smaller dataset, the number of transitive edges is comparable to the number of direct edges, so competitors' performance are closer to BINDER, but still substantially inferior. It validates that BINDER does not rely much on the presence of transitive edges in the training data, whereas other competing methods do. For larger tree-like datasets like Shwartz Random and WordNet Nouns, there are generally far more transitive edges than direct edges, and for both the datasets BINDER's performance is around or above 99%. As we add more transitive edges to the training dataset, the competitors' results improve, yet BINDER maintains its superiority over those methods (with a smaller margin); this further validates that other methods rely on transitive edges during training, whereas BINDER does not. We show direct and transitive edge statistics for all datasets in Appendix E

## 4 Other Related Works

We have discussed works that perform hierarchical embedding in the Introduction section and compared our results with those in the Results section. In terms of order embedding in binary space, to our best knowledge, there are no previous works. The closest works using FCA are Rudolph (2007), which provided ways to encode a formal context's closure operators into NN, and Dürrschnabel et al. (2019) where authors introduced fca2vec similar to node2vec Grover & Leskovec (2016), which embeds existing FCA systems into real-valued vector spaces using NN method. None of these works are our competitor, as their embedding objective is to embed in 2 dimensional real space for visualization. Zhang & Saab (2021) takes a $d$-dimensional vector representation of $N$ entities and embed those $N$ objects into $k$-dimensional binary vector space $[-1, +1]^k$ for preserving Euclidean distance between a pair of entities. The objectives of these works are different than ours, since they all require existing embeddings, while BINDER creates embeddings from a given hierarchy.

## 5 Future Works and Conclusion

BINDER is the first work which uses binary vector for embedding concepts, so there are numerous scopes for building on top of this work. First, we like to explore more efficient combinatorial

Table 4: Link Prediction (Transitive Closure) Results   Balanced Acc(%) (dim)

| Model | Medical | Music | Shwartz Lex | Shwartz Random | Nouns |
|---|---|---|---|---|---|
| **Transitive Closure 0%** | | | | | |
| OE | 96.7 (10) | 91.2 (5) | 81.7 (10) | 78.2 (200) | 80.7 (200) |
| Poincaré | 88.9 (50) | 72.6 (5) | 78 (20) | 73.5 (5) | 62.0 (100) |
| HEC | 86.9 (100) | 73.8 (100) | 79 (10) | 73.3 (5) | 70.2 (200) |
| T-Box | 70.4 (100) | 78.0 (100) | 74.6 (10) | 71.4 (100) | 70.8 (100) |
| BINDER | **99.4** (100) | **97.5** (100) | **99.7** (100) | **99.5** (100) | **98.4** (120) |
| **Transitive Closure 10%** | | | | | |
| OE | 96.3 (10) | 94.5 (10) | 85.4 (5) | 84.4 (5) | 90.1 (5) |
| Poincaré | 91.6 (10) | 78.5 (20) | 93.8 (50) | 87.4 (5) | 71.6 (5) |
| HEC | 91.9 (50) | 82 (50) | 95.2 (100) | 91.4 (50) | 97.3 (200) |
| T-Box | 80.0 (100) | 80.0 (100) | 74.4 (100) | 74.4 (100) | 80.1 (25) |
| BINDER | **99.4** (100) | **96.5** (100) | **100** (100) | **99.9** (100) | **99.7** (120) |
| **Transitive Closure 25%** | | | | | |
| OE | 97.4 (20) | 95 (5) | 88.6 (10) | 84.6 (50) | 93.3 (10) |
| Poincaré | 91.3 (5) | 82 (200) | 94.2 (100) | 91.5 (100) | 74.2 (10) |
| HEC | 93.9 (100) | 86.1 (100) | 96.2 (100) | 93 (10) | 97.9 (100) |
| T-Box | 80.8 (100) | 80.3 (100) | 74.6 (25) | 74.6 (100) | 86.8 (25) |
| BINDER | **99.1** (100) | **97.3** (100) | **100** (100) | **99.9** (100) | **99.7** (120) |
| **Transitive Closure 50%** | | | | | |
| OE | 97 (10) | 95.8 (50) | 90.3 (10) | 88.3 (10) | 95.6 (10) |
| Poincaré | 91.8 (5) | 85.6 (10) | 94.7 (50) | 90.9 (10) | 75.7 (5) |
| HEC | 94.5 (100) | 88.4 (5) | 96.5 (50) | 93.2 (100) | 98.4 (50) |
| T-Box | 83.8 (100) | 84.3 (100) | 74.4 (50) | 74.5 (100) | 91.2 (25) |
| BINDER | **99.7** (100) | **98** (100) | **100** (100) | **99.9** (100) | **99.9** (120) |

Table 5: Distribution of BINDER Results   Acc ($\mu \pm \sigma$)%

| Task | Medical | Music | Shwartz Lex | Shwartz Random | Nouns |
|---|---|---|---|---|---|
| Recon (100% TC) | $99.9 \pm 0.01$ | $99.9 \pm 0.01$ | $99.9 \pm 0.01$ | $99.9 \pm 0.1$ | $99.8 \pm 0.04$ |
| Pred (0% TC) | $98.9 \pm 0.3$ | $94 \pm 0.6$ | $99.6 \pm 0.1$ | $99.4 \pm 0.05$ | $98.1 \pm 0.2$ |
| Pred (10% TC) | $98.8 \pm 0.5$ | $94.7 \pm 1$ | $99.9 \pm 0.04$ | $99.8 \pm 0.1$ | $99.5 \pm 0.2$ |
| Pred (25% TC) | $98.9 \pm 0.2$ | $95.3 \pm 1.5$ | $99.9 \pm 0.03$ | $99.9 \pm 0.02$ | $99.6 \pm 0.1$ |
| Pred (50% TC) | $99.5 \pm 0.2$ | $94.9 \pm 1.6$ | $99.96 \pm 0.03$ | $99.9 \pm 0.02$ | $99.9 \pm 0.04$ |

TC = Transitive Closure

optimization algorithms for improving BINDER's learning algorithm by using various well-known CSP heuristics. Binder's Loss function can also be extended with a node similarity expression (dot product between node's attribute vectors, if present), which we plan to do next. It can also be extended to consider sibling similarity. In terms of limitation, BINDER is a transductive model, i.e., a concept must appear in the training data for its embedding to be learnt, but this limitation is also shared by all the existing hierarchical embedding models, including order-embedding, hyperbolic cone embedding, and Poincaré embedding. Yet, BINDER is better than the competitors as it can generate embedding for unseen concepts by using logical functions over existing concepts, which the competitors cannot do. Another future work can be to make BINDER inductive over any unseen concept by imparting knowledge regarding the broadness (or narrowness) of a concept from large distributed language model, such as BERT, RoBERTa, or Glove. To conclude, in this work, we propose BINDER, a novel approach for order embedding using binary vectors. BINDER is ideal for finding representation of concepts exhibiting hypernym-hyponym relationship. Also, BINDER's binary vector based embedding is extensive as it allows obtaining embedding of other concepts which are logical derivatives of existing concepts. Experiments of five benchmark datasets show that BINDER is superior than the existing state-of-the-art order embedding methodologies.

## REPRODUCIBILITY

We have included our Python code for training BINDER using PyTorch, including five datasets automatically split into train, validation, and test as shown in the `data` folder therein. Due to the stochastic nature of BINDER, we included the best achieved results in Table 4 but also ran multiple experiments to show more of BINDER's distribution of results (Table 5). Despite starting in a determined state, BINDER gives extremely diverse outputs, due to the random flipping and the fact that Algorithm 2 is symmetric in the $d$ dimensions.

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

APPENDICES

## A    PROOF OF BINDER'S CONVERGENCE TO LOCAL OPTIMAL SOLUTION

BINDER's algorithm literally solves a combinatorial satisfiability task, a known NP-complete problem. By computing gradient and then utilizing the gradient in deciding bit flipping probability, it works as a gradient descent scheme in the combinatorial space to minimize the objective function defined in Equation 4. However, note that the bit flipping probability decreases gradually as the number of violated constraints decreases with each subsequent epochs. This gives BINDER's learning algorithm a flavor of local search with simulated annealing. The following theorems and lemmas are provided to establish the claim of BINDER's local optimality.

**Lemma 1.** *When bias $b_\ell = 0$, for any word $a$, if the $j$'th bit in the binary representation vector $\mathbf{a}$ is updated by* BINDER*'s probabilistic flipping (keeping the remaining bits the same), the loss function value decreases in the successive iteration.*

*Proof.* Let $S$ be the set of positive data instances $(a, b)$ where the first entity is the given $a$. $\theta$ is a $d$-dimensional embedding vector of $a$ and $L(\theta)$ is the component of loss function associated with $a$. Suppose in an iteration, we probabilistically flip bit $j$ of $\theta$. To compute this probability, BINDER computes the $\Delta_{\mathbf{a}_j} Loss_P$, which is $L(\theta) - L(\theta'_j)$ , where $\theta'_j$ is the same as $\theta$ except that the bit value of $j$'th position is different. (Recall that we define our gradient to be *positive* when flipping bit $j$ improves our model, thus *decreasing* the loss function.) Based on Eq. 5, this gradient value is $+1$ only for the case when a constraint $\mathbf{a}_j \to \mathbf{b}_j$ is violated (where $b$ is the other element in a training pair) i.e. $\mathbf{a}_j = 0$, but $\mathbf{b}_j = 1$ (see the 3rd column of Table II). Using Line 7 of Algorithm 2 for $b_\ell = 0$, this yields a positive flip probability ($tanh$ is asymmetric function), and with the flip the loss function value decreases by $K\alpha$ (through Eq. 2), where $0 \le K \le |S|$; here $K$ is the number of pairs in $S$ that violate implication constraint with $a$ in the left side. For the other three choices of $\mathbf{a}_j$ and $\mathbf{b}_j$, $(0, 0), (1, 0), (1, 1)$, the contribution to gradient value is 0 or $-1$, yielding zero flip probability. In all scenarios, the loss value decreases in the successive iteration. □

**Lemma 2.** *When bias $b_\ell = 0$, for any word $b$, if the $j$'th bit in the binary representation vector of $b$ is updated by* BINDER*'s probabilistic flipping (keeping the remaining bits the same), the loss function value decreases in the successive iteration.*

*Proof.* The proof is identical to the proof of Lemma 1, except that we use gradient value in Eq. 6 instead of Eq. 5. In this case also when only one position of $b$'s embedding vector is flipped probabilistically, the loss function value decreases. □

**Lemma 3.** *When bias $b_\ell = 0$, given a collection of negative data instances, say, $(a', b')$, if the $j$'th bit in the vectors of $a'$ or $b'$ independently (not simultaneously) is updated by* BINDER*'s probabilistic flipping (keeping the remaining bits same), the loss function value decreases or remains the same in the successive iteration.*

*Proof.* The proof is identical to proof of Lemma 1, except that we use gradient value in Eq. 7 (or Eq. 9) for the case of $a'$, and gradient value of Eq. 8 (or Eq. 10) for $b'$, and the loss function value decreases through Eq. 3. □

These proofs also apply if $r_\ell \alpha \ge b_\ell > 0$ and $r_\ell \beta \ge b_\ell > 0$. In that case, we can flip a bit with zero gradient. Such flips do not immediately increase or decrease the loss function; however, they can allow BINDER to improve from a weak local optimum. In our experiments, $r_\ell \alpha$ and $r_\ell \beta$ are much larger than $b_\ell$, so our algorithm prioritizes actual improvements over zero-gradient steps.

**Theorem 4.** *When bias $b_l = 0$, if Line 8 of Algorithm 2 is executed sequentially for each index value $j$ for each of the entities,* BINDER *reaches a local optimal solution considering a 1-hamming distance neighborhood.*

*Proof.* Using earlier Lemmas, each bit flipping in any embedding vector of any of the entities, either decreases the loss function or keeps it the same. When Line 8 of Algorithm 2 is executed sequentially for each index $j$ (only one change in one iteration) for each of the entities, the loss function value

monotonically decreases in each successive iteration, At a local optimal point, none of the single bit flip improves the value of loss function. Now, if the bias $b_l = 0$, for each entity, the probability of bit-flipping for each index is computed to be 0 (by Line 7 in Algorithm 2), embedding of none of the entities changes any further and BINDER reaches a local optimal solution considering a 1-hamming distance neighborhood. In other words, for an entity $a$, considering that all the entity embedding is fixed, if we change any single bit of $a$'s embedding, the original embedding of $a$ is guaranteed to be at least as good as the changed embedding. □

When we change only one bit at a time keeping everything else the same (as in the proof), our optimization algorithm becomes a greedy hill climbing algorithm. However, this would make BINDER extremely slow to converge, and it may get stuck in a bad local optimal solution. Thus, we allow all bits to change simultaneously, so it behaves like gradient descent: Suppose $\theta$ is an embedding vector of an entity and $L(\theta)$ is the component of loss function associated with this entity. For minimizing $L(\theta)$, at each iteration, a hill climbing method would adjust a single element in $\theta$ to decrease $L(\theta)$; on the other hand, gradient descent will adjust all values in $\theta$ in each iteration by using $\theta^{new} = \theta^{old} - \alpha \Delta_\theta L(\theta^{old})$. During early iterations, BINDER works like gradient descent, but as iteration progresses, it behaves more like a hill climbing method as gradient values for most bit positions decrease, causing fewer bits to flip.

## B  PSEUDO-CODE

Pseudo-code of BINDER is shown in Algorithm 2. We initialize the embedding matrix with all 0's. The algorithm goes for at most $T$ epochs (for loop in Line 4-17), updating bit vectors of each vocabulary word in each iteration by flipping bits with probability based on gradient computed through Algorithm 1. The balanced-accuracy metric, defined as $\frac{1}{2} \left( \frac{TP}{TP+FN} + \frac{TN}{TN+FP} \right)$, is computed at the end of each epoch and best accuracy is recorded. We exit early if no improvement on validation data is seen over two consecutive windows of $\omega$ epochs, for user-specified $\omega$. The overall computational complexity is $O(ndT(|P| + |N|))$, for $n$ words and $d$ dimensions, which is linear in each variable.

---

**Algorithm 1** Gradient Computation

---

**Require:** Zero-one Embedding Matrix $B$ of size $n \times d$ initialized with all 0; positive Is-A relation set $P = \{(a^i, b^i)\}_{i=1}^m$; negative set $N = \{(a'^i, b'^i)\}_{i=1}^{m'}$; positive and negative sample weights $\alpha, \beta$

1: $\Delta^+ \leftarrow$ zero matrix, same size as $B$
2: $\Delta^- \leftarrow$ zero matrix, same size as $B$
3: **for** $(a, b) \in P$ **do**  $\qquad\qquad\qquad\qquad\qquad\qquad\qquad\qquad$ ▷ $*$ is element-wise product
4: $\qquad \Delta^+[a, :] \leftarrow \Delta^+[a, :] + B[b, :] * (1 - 2B[a, :])$
5: $\qquad \Delta^+[b, :] \leftarrow \Delta^+[b, :] + (1 - B[a, :]) * (2B[b, :] - 1)$
6: **end for**
7: **for** $(a', b') \in N$ **do**
8: $\qquad \mathbf{G} \leftarrow B[b', :] * (1 - B[a', :])$  $\qquad\qquad\qquad\qquad\qquad$ ▷ "good" bit pairs (a vector)
9: $\qquad$ **if** $\sum_j \mathbf{G}_j = 0$ **then**  $\qquad\qquad\qquad\qquad\qquad\qquad$ ▷ false positive, flip something
10: $\qquad\qquad \Delta^-[a', :] \leftarrow \Delta^-[a', :] + B[a', :] * B[b', :]$
11: $\qquad\qquad \Delta^-[b', :] \leftarrow \Delta^-[b', :] + (1 - B[a', :]) * (1 - B[b', :])$
12: $\qquad$ **else if** $\sum_j \mathbf{G}_j = 1$ **then**  $\qquad\qquad\qquad\qquad\qquad$ ▷ close to being wrong, so protect
13: $\qquad\qquad \Delta^-[a', :] \leftarrow \Delta^-[a', :] - \mathbf{G}$  $\qquad\qquad\qquad$ ▷ note only one element of $\mathbf{G}$ is 1
14: $\qquad\qquad \Delta^-[b', :] \leftarrow \Delta^-[b', :] - \mathbf{G}$
15: $\qquad$ **end if**
16: **end for**
17: **return** $\Delta := \alpha \Delta^+ + \beta \Delta^-$

---

## C  TRAINING SETUP AND HYPERPARAMETER TUNING

For BINDER, we learn a $d$-bit array for each concept in the hierarchy. For all tasks, we train BINDER for 10000 epochs, where each epoch considers the full batch for training. We tune our hyperparameters: dimension $d$, positive and negative sample weights $\alpha$, $\beta$, negative sample multiplier $n^-$, and the learning rate and bias $r_\ell, b_\ell$ manually by running separate experiments for each dataset. We

---

**Algorithm 2** Training Algorithm

---

**Require:** Word list $W = (w_1, \ldots, w_n)$; Dimension $d$; Positive training set $P = \{(a^i, b^i)\}_{i=1}^m$; validation sets $VP, VN$; gradient weights $\alpha, \beta$, learning params $r_\ell, b_\ell$, negative sample multiplier $n^-$ (must be even); maximum epochs $T$, early stop width $\omega$

1: $B \leftarrow$ zero matrix of size $|W| \times d$
2: $Acc \leftarrow$ empty list
3: $(BestEmbedding, BestAcc) \leftarrow (B, 0)$
4: **for** $t = 1$ to $T$ **do**
5:      $N \leftarrow$ negative samples (Section 2.3)
6:      $\Delta \leftarrow$ gradient from Algorithm 1
7:      $X \leftarrow \max\left\{0, \frac{1}{2}\tanh(2(r_\ell\Delta + b_\ell))\right\}$                      ▷ flip probabilities
8:      Flip each bit $B[w, j]$ with (independent) probability $X[w, j]$
9:      $acc \leftarrow$ BalancedAccuracy(Evaluate($B, VP, VN$))
10:      **if** $acc > BestAcc$ **then**
11:          $(BestEmbedding, BestAcc) \leftarrow (B, acc)$
12:      **end if**
13:      Append $acc$ to list $Acc$
14:      **if** mean(last $2\omega$ elements of $Acc$) $\geq$ mean(last $\omega$ elements of $Acc$) **then**
15:          Exit Loop                      ▷ Early Exit Criterion if no improvement
16:      **end if**
17: **end for**
18: **return** $BestEmbedding$

---

find that the optimal learning rate $r_\ell$ and learning bias $b_\ell$ are 0.008 and 0.01 respectively for all data sets and tasks. The learning bias $b_\ell = 0.01$ means that bits whose gradient was exactly neutral had a 1% chance of flipping. We fix $\beta$ at 10 and tuned $\alpha$; we always find that $\alpha \leq \beta$ gives far too many false negatives. By compressing the expressiveness of the model, we force the binary embeddings to make "decisions" about which attributes to merge, thus increasing its predictive power. For reconstruction task we need more bits to increase the capacity of our model to better reconstruct training edges. Optimal $(\text{bits}, \alpha, n^-)$ for reconstruction task on Medical, Music, Shwartz Lex and Shwartz Random datasets is (50, 30, 32), and for WordNet Nouns dataset it is (80, 15, 8). For link prediction transitive closure experiment on Medical, Music, Shwartz Lex and Shwartz Random datasets with transitive closure 0% and 10%: (100, 25000, 12) and for 25% and 50%: (100, 50000, 12). Optimal $(\text{bits}, \alpha, n^-)$ for link prediction transitive closure task on WordNet Nouns dataset with all transitive closure configurations (0%, 10%, 25%, 50%) is (120, 25000, 12). We use these hyper parameters to obtain results of Table 3, 4 and 5. For the competing models, except T-Box, we exhaustively tuned dimensions $d = 5, 10, 20, 50, 100, 200$ keeping other hyperparmeters similar to original papers. Since T-Box requires 2*d dimension for its representation, to be fair with other models, we tuned T-Box for dimensions $d = 2, 5, 10, 20, 50, 100$ keeping other hyperparmeters similar to original paper implementation. All the models were run on a Tesla V100 GPU.

## D    CASE STUDY

We present a case-study experiment, which will provide the reader a sketch of BINDER's embedding results. For this we run our model with 8 bits on the toy lattice from Vendrov et al. (2015). Because the lattice is very small, it is possible for BINDER to achieve perfect accuracy. Figure 1 shows the final embedding. Given the embeddings for `boy`, `person`, and `city`, a human can determine that, according to the model, `boy` is-a `person` but not is-a `city`. In theory, each bit can correspond to some "attribute" of each object, where all attributes are passed down to hyponyms. This can help to build an explainable embedding. For instance, the second circle clearly denotes an attribute which could be named *has-life*. Sometimes, however, bits are used with different meanings in different words: the right-most bit is 1 on `man`, `girl`, and `SanJuan`. This is partly because the toy lattice is very sparse, with only two sub-components of `city` compared to ten of `livingThing`, and `adult` and `child` were not included in the lattice, as they are in WordNet.

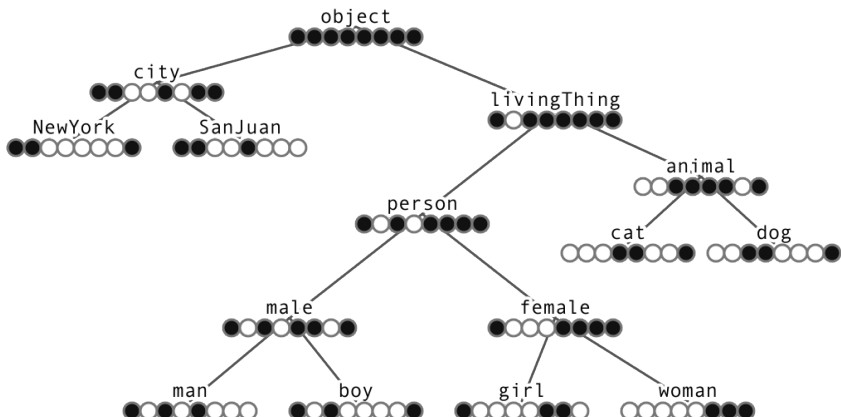

Figure 1: Visual representation of toy dataset results. White circles represent 1 and black circles 0.

Table 6: Edge distribution for all datasets

| Dataset | Edge Counts | | |
|---|---|---|---|
| | Direct Edge | Transitive (Indirect) Edge | Full Transitive Closure (Direct + Transitive) |
| Medical | 2616 | 1692 | 4308 |
| Music | 3920 | 2608 | 6528 |
| Shwartz Lex | 5566 | 7940 | 13506 |
| Shwartz Random | 13740 | 42437 | 56177 |
| WordNet Nouns | 84363 | 576764 | 661127 |

## E  DATASET STATISTICS

We have explained before that since BINDER does not rely on transitive edges for link prediction, unlike its competitors, BINDER performance is superior compared to its competitors at 0% transitive closure. The difference margin with the competitors are larger for large datasets where transitive edge percentages are significantly higher compared to direct edge percentages, as shown in Table 6 and figure 2. Hence it proves the superiority of BINDER embedding.

## F  MORE EXPERIMENTAL RESULTS

### F.1  ABLATION STUDY

For an ablation study, we observe the effect on model accuracy on the validation data by removing $\beta$ and $b_\ell$ separately while keeping other parameters at best value. For Nouns data set on reconstruction task, setting $\beta = 0$ (i.e. ignoring the negative samples) gives accuracy 76% after 500 iterations. If we set $b_\ell$ to 0 then we don't see any significant effect on accuracy. For the 0% transitive closure prediction task on Nouns data set, when we set $\beta = 0$ accuracy saturates at 86% after 800 iterations. If we set $b_\ell$ to 0 then we see accuracy drops by several percentages from the best result.

### F.2  JUSTIFICATION OF BALANCED ACCURACY METRIC OVER F1-SCORE

For classification over imbalance data, F1 measure is a good metric; however some argue (and we agree with them) that balanced accuracy (BA) is actually a better metric, which is defined as the average of positive class $recall_+ = \frac{TP}{TP+FN}$, and negative class $recall_- = \frac{TN}{TN+FP}$. Balanced accuracy is a better metric than F1-score for imbalanced datasets, because F1-score does not care about how many true negatives are being classified. F1-score uses precision and recall, which together use only three entries of the confusion matrix (TP, FP, FN); on the other hand, balanced accuracy uses all four entries of the confusion matrix. For an example, say a dataset has 1000 negative and 10 positive examples. If the model predicts there are 15 positive ($TP = 5, FP = 10$), and predicts the rest as negative ($TN = 990, FN = 5$), we get $Precision = \frac{5}{15} \approx 0.33$, and $Recall = \frac{5}{10} = 0.5$ it yields $F_1 = 0.4$. The model does not get much credit for correctly predicting 990 out of 1000

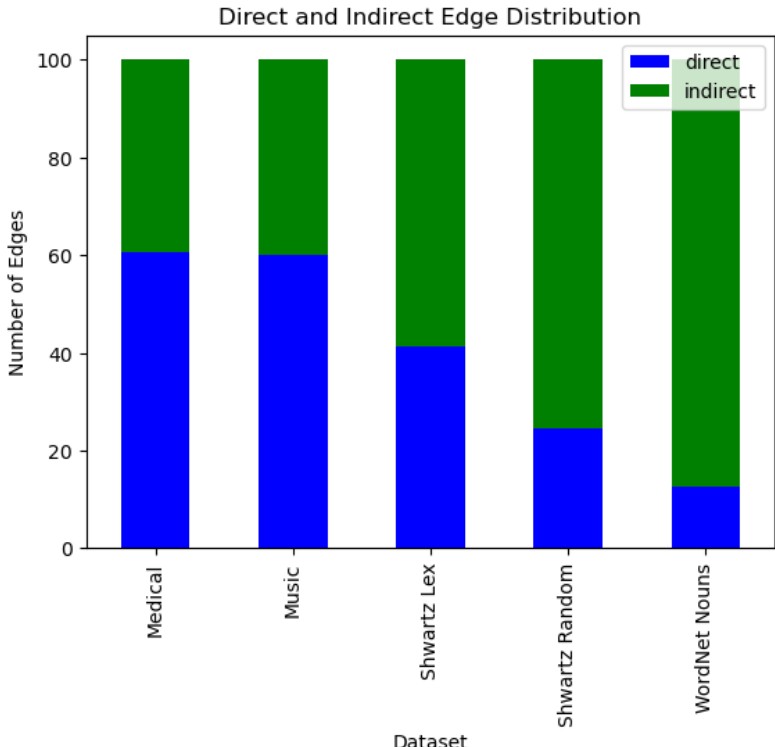

Figure 2: Distribution of Direct and Transitive (Indirect) edges for all datasets.

examples as negative. However, the balanced accuracy is $\frac{1}{2} * \left( \frac{5}{10} + \frac{990}{1000} \right) = 0.745$, which provides somewhat a more realistic picture.

Additionally, balanced accuracy does not depend on the ratio of positive to negative edges, which is important because our datasets have different negative edge ratios: Medical has about 1400 entities and 4300 edges, for a negative ratio of about 450, while the ratio for Nouns is about 9000, and evaluating the entire negative edge set is impractical. In practice, we chose a 10:1 ratio of negative to positive test cases, but this choice was arbitrary and made because Nickel & Kiela (2017) used the same ratio. Balanced accuracy is unaffected by these arbitrary decisions, provided the samples are large enough to avoid statistical error.

We actually used an F1 measure with the 10:1 ratio, but found it to be very harsh for the competitors. In the Table below, we show F1-score results for the four transitive link prediction experiments. As can be seen BINDER's F1-scores (bold numbers) are significantly better than all other methods on all datasets. The competitor methods suffer severely due their poor precision. We have reported F1-score for all our experiments in Tables 7, 8 and 9.

### F.3 BINDER: RESULTS FURTHER DISCUSSION

For reconstruction task, OE achieves better performance than BINDER on two smaller datasets (Edges: Medical: 4.3k, Music: 6.5k) and achieves equal performance on mid-sized datasets (Edges: Shwartz Lex: 13.5k, Shwartz Random: 56.2k) for fewer dimensions. For smaller datasets, OE achieves 100% accuracy by using $d = 10$ and 20, but BINDER achieves 99.9% accuracy by using 50 bits. Based on this observation, one can conclude that BINDER does not show superiority over OE. However, we argue that BINDER still wins because it uses bits, whereas other methods operates in real number domain. We need at least 4 bytes (32 bits) to represent a real number. So, the claim that OE is using fewer dimension is untrue, because OE is using $d \geq 10$ ($10 \times 32 = 320$ bits), whereas BINDER is using only $d = 50$ bits. If BINDER is allowed only 50 bits, for a fair comparison other methods should be allowed $\lceil 50/32 \rceil = 2$ dimensions. From our experiments,

Table 7: Reconstruction Results   F1-score(%) (dim)

| Model | Medical
entities = 1.4k
edges = 4.3k | Music
entities = 1k
edges = 6.5k | Shwartz
Lex
entities = 5.8k
edges = 13.5k | Shwartz
Random
entities = 13.2k
edges = 56.2k | WordNet
Nouns
entities = 82k
edges = 743k |
|---|---|---|---|---|---|
| OE | **100** (20) | **100** (20) | 100 (20) | 100 (50) | 97.5 (200) |
| Poincaré | 61.0 (100) | 45.0 (50) | 40.1 (10) | 28.6 (100) | 97.2 (50) |
| HEC | 87.5 (100) | 73.2 (100) | 97.7 (20) | 88.6 (10) | 91.3 (100) |
| T-Box | 100 (25) | 100 (50) | 100 (25) | 100 (50) | 99.9 (50) |
| BINDER | 99.9 (50) | 99.9 (50) | **100** (50)* | **100** (50)* | **99.9** (80)* |

*For Shwartz Lex and Random dataset, BINDER dimension is higher compared to OE but considering space complexity (1 bit vs 4 bytes) for each dimension we conclude BINDER as the wining model.

Table 8: Link Prediction (Transitive Closure) Results   F1-score(%) (dim)

| Model | Medical | Music | Shwartz
Lex | Shwartz
Random | Nouns |
|---|---|---|---|---|---|
| **Transitive Closure 0%** | | | | | |
| OE | 83.1 (10) | 74.8 (10) | 46.7 (10) | 42.1 (50) | 48.9 (20) |
| Poincaré | 44.0 (100) | 27.5 (50) | 28.8 (5) | 31.9 (5) | 33.1 (200) |
| HEC | 58.3 (100) | 38.2 (20) | 41.2 (50) | 32.3 (5) | 39.6 (200) |
| T-Box | 29.8 (50) | 29.5 (50) | 25.5 (100) | 22.8 (50) | 25.7 (100) |
| BINDER | **96.6** (100) | **87.8** (100) | **98.4** (100) | **97.5** (100) | **91.7** (120) |
| **Transitive Closure 10%** | | | | | |
| OE | 87.3 (100) | 81.8 (20) | 56.7 (5) | 51.6 (5) | 62.1 (5) |
| Poincaré | 55.2 (50) | 30.0 (10) | 22.6 (5) | 27.8 (200) | 35.6 (10) |
| HEC | 71.5 (50) | 51.6 (50) | 81.1 (200) | 66.8 (50) | 87.2 (200) |
| T-Box | 38.7 (100) | 35.2 (100) | 28.3 (100) | 28.5 (100) | 35.6 (100) |
| BINDER | **99.4** (100) | **83.9** (100) | **100** (100) | **99.9** (100) | **99.2** (120) |
| **Transitive Closure 25%** | | | | | |
| OE | 87.3 (10) | 83.2 (20) | 55.9 (10) | 51.1 (10) | 71.2 (10) |
| Poincaré | 54.4 (10) | 33.2 (20) | 22.7 (20) | 23.4 (10) | 39.9 (50) |
| HEC | 78.9 (100) | 58.4 (5) | 85.9 (20) | 74.1 (50) | 91.2 (200) |
| T-Box | 39.6 (100) | 35.4 (100) | 28.4 (100) | 28.5 (100) | 45.6 (100) |
| BINDER | **99.1** (100) | **87.0** (100) | **100** (100) | **99.9** (100) | **98.5** (120) |
| **Transitive Closure 50%** | | | | | |
| OE | 92.0 (50) | 87.9 (10) | 53.3 (5) | 61.6 (10) | 80.7 (10) |
| Poincaré | 62.1 (50) | 32.4 (200) | 24.4 (100) | 23.6 (20) | 43.0 (200) |
| HEC | 87.2 (200) | 68.9 (5) | 83.1 (50) | 77.3 (100) | 95.3 (50) |
| T-Box | 48.8 (100) | 42.5 (100) | 28.1 (100) | 28.5 (50) | 57.9 (100) |
| BINDER | **99.7** (100) | **90.0** (100) | **100** (100) | **99.9** (100) | **99.6** (120) |

we observed that even with $d = 5$ ($5 \times 32 = 160$ bits) and $d = 10$ (320 bits), OE achieves 99.4% and 98.6% accuracy respectively for Medical and Music, which is poorer than BINDER. Most importantly, on the largest dataset (Nouns), which has 100 times more edges than the smaller datasets, BINDER achieves 99.7% accuracy with 100 bits, whereas OE achieves 96.7% accuracy by using 200 dimensions, or $200 \times 32 = 6400$ bits.

To summarize, OE uses the smallest dimension of all the competitors because it uses real-number space which is less constrained, whereas hyperbolic space is more constrained, and box embedding requires two vectors per dimension. Comparing to BINDER, OE actually uses more memory, as OE uses real space and BINDER uses binary vectors.

Table 9: Distribution of BINDER Results   F1-score $(\mu \pm \sigma)\%$

| Task | Medical | Music | Shwartz Lex | Shwartz Random | Nouns |
|---|---|---|---|---|---|
| Recon (100% TC) | $99.9 \pm 0.03$ | $99.8 \pm 0.04$ | $99.9 \pm 0.11$ | $99.9 \pm 0.07$ | $99.8 \pm 0.04$ |
| Pred (0% TC) | $93.9 \pm 2.4$ | $78.2 \pm 5.5$ | $97.7 \pm 0.5$ | $96.9 \pm 0.3$ | $98.1 \pm 0.2$ |
| Pred (10% TC) | $93.8 \pm 2.5$ | $77.4 \pm 3.7$ | $99.4 \pm 0.3$ | $99.1 \pm 0.1$ | $97.8 \pm 1.1$ |
| Pred (25% TC) | $94.1 \pm 0.2$ | $79.6 \pm 5.8$ | $99.4 \pm 0.3$ | $99.5 \pm 0.1$ | $97.7 \pm 0.9$ |
| Pred (50% TC) | $97.1 \pm 0.9$ | $78.4 \pm 6.5$ | $99.8 \pm 0.2$ | $99.6 \pm 0.1$ | $99.4 \pm 0.3$ |

TC = Transitive Closure

## F.4 BINDER: MODEL CONVERGENCE RESULTS

We run our models with a large number of iterations to maximize the accuracy. Although the model attains high accuracy very quickly, it continues to improve steadily for reconstruction task, as shown in the first graph in Figure 3. The $0\%$ transitive closure prediction task accuracy saturates at around 1000 iterations and then start decreasing, as shown in the second graph in Figure 3.

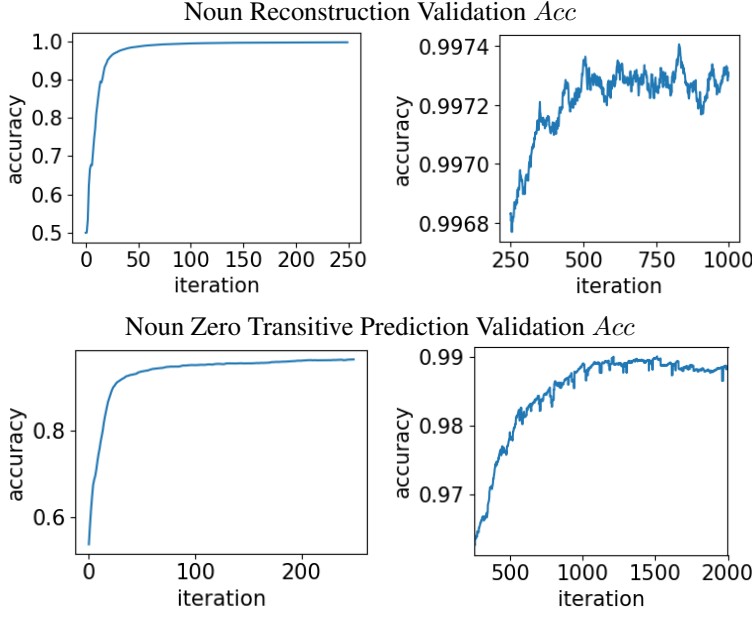

Figure 3: Graph of validation $Acc$ score of the two types of Noun experiments, for the first 250 iterations (left) and the last 750 and 1750 (right).

## F.5 BINDER: RESOURCE CONSUMPTION RESULTS

One of the main advantages of using binary vectors is their efficiency compared to floating-point vectors. The algorithms for BINDER are fast since BINDER uses only basic integer arithmetic except for the `tanh` function in the flip probability. On Nouns dataset the Reconstruction task takes 22m 50s for 1000 iterations and the $0\%$ transitive closure task takes 2m 09s for 2000 iterations. Furthermore, the final representation of the concepts using BINDER are binary vectors; the storage required for $n$ words and $d$ dimensions is $\frac{dn}{8}$ bytes instead of Order Embedding's $4dn$ bytes if 32-bit floating point values are used. We want to add the following table where we consider WordNet Nouns dataset and calculate the size of final embedding of different models for d=100 in table 10

Table 10: Space complexity for all models

| Model | Storage |
|---|---|
| OE | 34.2 MB |
| Hyperbolic methods | 34.2 MB |
| T-Box | 67.0 MB |
| BINDER | 2.36 MB |

# G    FURTHER JUSTIFICATION OF BINDER'S ALGORITHM

## G.1    COMPARISON OF BINDER WITH SPARSE ADJACENCY LIST (SAL)

Sparse adjacency list (SAL) does not provide fixed-size vector embedding of entities, but BINDER's bit-vector representation provides that. SAL does not capture order between entities, but BINDER's bit-vector provides that. SAL only captures the edges of the relation, but BINDER's bit-vector is an order-embedding, which represents nodes as vectors which are transferable to other subsequent knowledge discovery task. For the Noun dataset, which has 743K edges and 82K vertices, BINDER's bit vector will take $82\,\text{kB} \cdot 100/8 = 1025\,\text{kB}$, whereas a sparse adjacency list will take at least $(743\text{k} + 82\text{k}) * 4 = 3300$ Kbytes (considering 4 bytes for integer).

## G.2    COMPARISON OF BINDER'S RANDOMIZED ALGORITHM WITH DETERMINISTIC ALGORITHM FOR GENERATING BIT-VECTOR EMBEDDINGS

A deterministic algorithm offers no learning, it is simply memorizing the edges. It can only be used when an entire minimal set of edges of the DAG is given. But, one cannot expect that all the edges between entity pairs are already known/given in the training data. If that is the case, no learning or embedding is needed and a fixed deterministic method can be used.

The advantage of BINDER is that it has learning capability. It assign bit-vectors so that it can infer missing edges. In other words, if we remove some direct edges in the training data, BINDER will still be able to embed entities reasonably. We performed link prediction experiments to prove this claim on Mammals and Nouns dataset; this setup is identical to that of Vendrov et al. (2015). For Mammals dataset (6.5k edges) and Nouns dataset (743k edges) we randomly take out 300 and 3500 edges[2] respectively, which may or may not be direct edges, to construct positive test data. We create negative test data by corrupting positive pairs. Results from this experiment are reported in table 11. For Nouns, Vendrov et al. (2015) reported an accuracy of 90.6% in their work, which is worse than BINDER's 93.9%.

Table 11: Results of Vendrov et al. (2015) style Link Prediction experiment

| Model | Dataset | | | |
|---|---|---|---|---|
| | Mammals | | Nouns | |
| | Acc(%) | F1-score(%) | Acc(%) | F1-score(%) |
| BINDER | $94.2 \pm 1.1$ | $94.1 \pm 1.1$ | $93.9 \pm 0.4$ | $93.6 \pm 0.5$ |

---

[2]Vendrov et al. (2015) removed 4000 edges from their Nouns dataset. However, they appear to have used a different version of WordNet Nouns with about 838,000 edges, and so we remove fewer edges to maintain approximate proportion.

