# OpenReview forum: "Binder: Hierarchical Concept Representation through Order Embedding of Binary Vectors"
_ICLR.cc/2024/Conference — Submitted to ICLR 2024_

### Official Review · Reviewer_wKPf · 2023-10-21

**Soundness:** 4 excellent
**Presentation:** 1 poor
**Contribution:** 2 fair
**Rating:** 5
**Confidence:** 3

**Summary:**

The authors proposed a novel embedding method to represent data points that have partial order relation. The author's idea is to map those data points to bit sequences, where the inequality of the values of two sequences corresponding to the original partial order relation.

**Strengths:**

1. The original idea of using bit sequences is novel as far as I know, simple, easy to understand, and intuitive to some extent.
1. The authors successfully associate the proposed method with the existing order embedding, which helps the authors' understanding.
1. The algorithm's explanation also maintains some intuition.
1. The algorithm has strong advantage on the space computational complexity.
1. Overall, the technical parts of the paper are well-written.

**Weaknesses:**

Overall, the presentation of the paper needs essential refinement. The current version's presentation degrades the paper's quality although the research idea itself is nice and impressing to me.

1. From the introduction, the ultimate motivation of the work is not very clear. For continuous space embedding case, we could use them for visualization or we could input the representations to another machine learning architecture, such as neural network. However, we have no clear idea how we can use the obtained binary embedding in applications. If we just want to do link prediction or reconstruction, we do not need to stick to embedding-based methods.

1. As a starting motivation of the research, the paper criticizes hyperbolic embedding, pointing out that "learning in hyperbolic space is challenging because optimization algorithms, like gradient descent and its variants, are not well studied for hyperbolic space." Indeed, the gradient descent methods on hyperbolic space have been well-studied theoretically, e.g., [A-E]. Although the convergence to the global optimum cannot be guaranteed, as not in Euclidean space, but they are not by far worse than the author's theoretical guarantee on the proposed algorithm. The author mentioned that the problem is a NP-complete problem as a decision problem, but it is not a practically positive result unless P=NP. In this sense, the current draft gives readers impression that the author has not solved the original motivation. If it is difficult to provide a theoretical guarantee of the proposed algorithm, the author should criticize the hyperbolic embedding in another way.

1. This item is about another important motivation of the paper, "logical operation." The explanation regarding the logical operation on the binary representations does not seem correct. The logical "not" operator does not seem to work like the semantic "not." Assume that "living thing" is [0, 0], "cat" is [0, 1], and "dog" is [1, 0]. This does not self-contradict since a cat is a living thing and a dog is a living thing, too. Let's apply the logical "not" to the living thing. According to your explanation, "not living thing" is [1, 1]. Now, according to the rule, we conclude that "a not living thing is a cat", and "a not living thing is a dog." This is obviously wrong. Hence, the proposed boolean representations are not intuitive as the authors claim.

1. Citation does not include which year it is published, which makes it extremely difficult to see the flow of the existing methods.

1. The page limitation is violated.

1. As I discuss in the Questions section, the advantages of the proposed methods do not seem completely stated in the current draft.

[A] Zhang, Hongyi, and Suvrit Sra. "First-order methods for geodesically convex optimization." In Conference on Learning Theory, pp. 1617-1638. PMLR, 2016.

[B] Zhang, Hongyi, Sashank J Reddi, and Suvrit Sra. "Riemannian SVRG: Fast stochastic optimization on Riemannian manifolds." Advances in Neural Information Processing Systems 29 (2016).

[C] Liu, Yuanyuan, Fanhua Shang, James Cheng, Hong Cheng, and Licheng Jiao. "Accelerated first-order methods for geodesically convex optimization on Riemannian manifolds." Advances in Neural Information Processing Systems 30 (2017).

[D] Zhou, Pan, Xiao-Tong Yuan, and Jiashi Feng. "Faster first-order methods for stochastic non-convex optimization on Riemannian manifolds." In The 22nd International Conference on Artificial Intelligence and Statistics, pp. 138-147. PMLR, 2019.

[E] Bécigneul, Gary, and Octavian-Eugen Ganea. "Riemannian adaptive optimization methods." ICLR 2019.

**Questions:**

- Why did you not show the memory (RAM) to store the representation in the numerical experiments? I thought you could emphasize the advantage of the proposed method clearer this way, since the proposed method's type is boolean, and existing method's types are float or double.
- Why did you not show the order of time complexity of the proposed algorithm? If I understand it correctly, each step's time and space complexity is linear to the dimension, which seems to be an advantage of the proposed algorithm.

---

> ### Author Response · Authors · 2023-11-20
> **Response to Reviewer wKPf**
>
> Thank you for your time to review our paper. Below is our response.
>
> Weakness comments:
>
> WK1. From the introduction, the ultimate motivation of the work is not very clear. For continuous space embedding case, we could use them for visualization or we could input the representations to another machine learning architecture, such as neural network. However, we have no clear idea how we can use the obtained binary embedding in applications.
>
> A: The ultimate motivation is the same as it was for the competing works that we compared against, of course with an objective to do it with better performance that the competitors. From experimental results we have validated that we were successful in our objective. To reiterate: We solve the following task: *From a collection of is-a relations between a pair of entities, obtain embedding of the entities such that embedding vectors geometrically capture the order imposed through the is-a relation.*
>
> The only difference from our competitors is that we are using binary bits for embedding and they are using real number vectors. However, binary bits are compact, but not limiting in any way. We can use binary bits for visualization and can also feed them in a neural network for subsequent knowledge discovery tasks, identically as one would use a real-vector for those purposes. A binary d-dimensional vector is also a point in the real-space, so any system that takes a real-vector should be able to take those binary vectors as input.
>
> WK2: As a starting motivation of the research, the paper criticizes hyperbolic embedding, pointing out that "learning in hyperbolic space is challenging ...  are not well studied for hyperbolic space." Indeed, the gradient descent methods on hyperbolic space have been well-studied theoretically, e.g., [A-E]. Although the convergence
>   .... hyperbolic embedding in another way.
>
> A:  After reading the references that you have listed, we admit our lack of knowledge about the latest advances of ML in hyperbolic space. What we meant to say in our comment is that optimization in hyperbolic space is more challenging that real-space and this is how we want to change the above sentence in a revised manuscript. We thank you for reading our paper including the Appendix to learn about BINDER's convergence claim, and we agree with your comment entirely. Nevertheless, without criticizing competitors framework, we like to point you towards the superior experimental results of BINDER over its competitors.
>
> WK3: This item is about another important motivation of the paper, "logical operation." The logical "not" operator does not seem to work like the semantic "not." Assume that "living thing" is [0, 0], "cat" is [0, 1], and "dog" is [1, 0]. This does not self-contradict since a cat is a living thing and a dog is a living thing, too. Let's apply the logical "not" to the living thing. According to your explanation, "not living thing" is [1, 1]. Now, according to the rule, we conclude that "a not living thing is a cat", and "a not living thing is a dog." This is obviously wrong. Hence, the proposed boolean representations are not intuitive as the authors claim.
>
> A: We refer you to the answer that we provided to Q4 of Reviewer VtjX as both your question and that question are similar. We only claim that we can obtain an embedding of an entity outside our domain by bit operation (bit operation when applied on a bit vector yield another bit vector, due to closeness property), but we cannot guaranty logical implication of an outside domain entity as those entity are not part of the training data. Going with your example: yes "not living thing" has a representation of [1, 1], and we stop right there, rather than making further implication, based on this embedding. We hypothesize, for real-life purpose, such embedding built by boolean operation will suffice for downstream knowledge discovery task, but validation of that claim is outside the scope of this paper. Please note, we clearly noted (sec. 5) that out method is transductive, just like SOTA methods proposed for this task.
>
> WK4: Citation does not include which year it is published.
>
> A: We are sorry about this and thank you very much for noticing this. We have fixed it already in our manuscript.
>
> WK5: The page limitation is violated.
>
> A. We are within the page limit (9 pages of main text). References, Appendix, and Reproducibilty does not count towards page limit.
>
> WK6: As I discuss in the Questions section, the advantages of the proposed methods do not seem completely stated in the current draft.
>
> A: There are three major advances: (1) A compact embedding method for is-a entities using binary bits; (2) A very efficient algorithm, for which the training time is about 10~20 minutes for graph having around 743K edges, some completing algorithms took about 10 hours in this dataset; (3) Most importantly, our method performs exceptionally well on link prediction task on 0% transitive closure dataset than the competitors.

---

> > ### Author Response · Authors · 2023-11-20
> > **Response to Reviewer wKPf (cont.)**
> >
> > Responses to Questions:
> >
> > Q1: Why did you not show the memory (RAM) to store the representation in the numerical experiments? I thought you could emphasize the advantage of the proposed method clearer this way, since the proposed method's type is boolean, and existing method's types are float or double.
> >
> > A: We had to compromise lot of content of this paper due to ICLR page limit, as you can see from our 5 pages of Appendix. In Appendix F.3, we mentioned that for $n$ entities and $d$ dimension, we only need $\dfrac{nd}{8}$ bytes, whereas competitors take $ndK$ bytes where $K$ is the number of bytes needed for a floating point (4 or 8). We will emphasize that in the main part of our revised manuscript.
> >
> > Q2: Why did you not show the order of time complexity of the proposed algorithm? If I understand it correctly, each step's time and space complexity is linear to the dimension, which seems to be an advantage of the proposed algorithm.
> >
> > A. Yes your understanding is correct. The complexity equation is provided in Appendix B along with Pseudo-Code. Quoting from our paper: "The overcall computational complexity is $O(ndT(|P|+|N|))$, for $n$ words, and $d$ dimensions, which is linear in each variable.". We will revise the manuscript to bring this line in the main part of the paper from the Appendix.

---

> > > ### Comment · Reviewer_wKPf · 2023-11-23
> > >
> > > Thank you for your detailed answer. They answer my questions perfectly.
> > >
> > > - **A for Q1**
> > >
> > > **My comments**: I acknowledge that your paper has many interesting contents and not easy to include all in the body text. However, since you stated many times that your method is compact in the Introduction section, you could prioritize showing compactness in the experiments to make your paper consistent.
> > >
> > > - **A for Q2**
> > >
> > > **My comments**: Again, since you mention the efficiency of your algorithm in the Introduction section, the author might want to include the evaluation in the body text.

---

> > ### Comment · Reviewer_wKPf · 2023-11-23
> >
> > First, I appreciate the author's sincere and detailed response. with updated manuscript
> >
> > - **A for WK1:**
> >
> >   **My comments**: You mentioned "*We solve the following task: From a collection of is-a relations between a pair of entities, obtain embedding of the entities such that embedding vectors geometrically capture the order imposed through the is-a relation.*"  If it is the case, the statement should come in the manuscript. Also, the geometric interpretation of the embedding by your method should be given in the body text. If I understand it correctly, converting your algebraic explanation of your method in the paragraph "In this work" to geometric interpretation is easy since the is-a relation corresponds to a cone in a vector space as well in your method, like an entail cone method. In any case, your motivation's description and method's description should be consistent.
> >
> > - **A for WK2:**
> >
> >   **My comments**: This is the most significant concern for me. If you propose the method without criticizing the existing method, readers cannot understand why you are trying to propose a new method. Even if you display your method's excellent results, readers cannot see why your method is excellent. I do not suggest the author stop discussing why your method is good.
> >
> > - **A for WK3:**
> >
> >   **My comments**: You mentioned "Going with your example: yes "not living thing" has a representation of [1, 1], and we stop right there, rather than making further implication, based on this embedding." I still do not see the reason why we can "stop right there," while your initial idea was to identify $x\_{i} \\le y\_{i}$ with $x_{i} \\Rightarrow y\_{i}$. You might want to simply omit the explanation about the reversing bits in the future presentation.
> >
> > - **A for WK4:**
> >
> >   **My comments**: I can't see the update in the system. In any case, I hope it will be fixed.
> >
> > - **A for WK5:**
> >
> >   **My comments**: It was my misunderstanding. I sincerely apologize for it.
> >
> > - **A for WK6:**
> >
> >   **My comments**: The answers here were not what I expected, but instead, the authors clearly answered my questions (Dear meta reviewers, see the post below), so it suffices.
> >
> > **Overall comments**: Overall, your answers for WK2 and 3 leave us with significant concerns, so I cannot raise my score. However, I acknowledge that this paper has promising potential and I strongly encourage the authors to improve the presentation to show its maximum potential in the future even if our decision is not positive to you.

---

> > > ### Author Response · Authors · 2023-11-23
> > > **Response to Reviewer WKPf**
> > >
> > > Thanks for your further comments. We have revised our manuscript and uploaded it. We firmly believe that we have addressed all your concerns in the revised manuscript, particularly regarding Wk2 and Wk3. We again thank you for your detailed review which helped to improve the quality of our work.
> > >
> > > We humbly request you to check our revised manuscript and appeal to reconsider increasing our score.

---

### Official Review · Reviewer_EAos · 2023-10-28

**Soundness:** 3 good
**Presentation:** 4 excellent
**Contribution:** 3 good
**Rating:** 6
**Confidence:** 4

**Summary:**

The paper introduces a novel approach, BINDER, for order embedding using binary vectors. BINDER aims to represent concepts with hypernym-hyponym relationships in a binary vector space, allowing for embeddings of both seen and unseen concepts. Experimental results demonstrate BINDER's superiority over existing order embedding methodologies.

**Strengths:**

1) Conceptual Simplicity: BINDER offers a novel and conceptually simple approach to hierarchical representation learning by using binary vectors. This simplicity is an advantage because it makes the method more interpretable and easier to understand compared to complex, black-box models.

2) Strong Performance in Reconstruction Task: BINDER consistently demonstrates excellent performance in the reconstruction task. This indicates its robust ability to learn embeddings that satisfy order constraints, which is a critical aspect of hierarchical representation learning.

3) Transitive Closure: BINDER's ability to predict hypernymy relations without relying heavily on transitive closure in the training data is a significant strength. This property suggests that the model can generalize effectively to unseen concepts and is not overly dependent on the availability of transitive edges.

4) Originality of Approach: BINDER introduces a unique approach to order embedding using binary vectors. This originality stems from its different perspective on hierarchical representation learning and adds to the diversity of methods in this field.

5) Potential for Extensions: The paper hints at possible extensions, such as incorporating node similarity expressions and considering sibling similarity. These extensions have the potential to enhance BINDER's capabilities and could pave the way for future research.

**Weaknesses:**

1) Generalization to Unseen Concepts: While BINDER claims to generate embeddings for unseen concepts by using logical functions over existing concepts, it would be beneficial to provide more detailed explanations and examples of how this generalization is achieved. A concrete illustration of how BINDER generates embeddings for unseen concepts could strengthen the paper.

2) Experimental Rigor: The paper mentions that BINDER is a randomized algorithm but provides results from the best run out of five. It would be helpful to include more detailed information on the variability observed in these runs, such as mean and standard deviation. A discussion of the algorithm's sensitivity to random initialization would also be insightful.

3) Hyperparameter Sensitivity: The paper discusses hyperparameters like the learning rate and bias but does not delve into their sensitivity analysis. A study on how these hyperparameters affect BINDER's performance and convergence would provide a better understanding of its behavior.

4) Comparative Discussion: While BINDER's strengths are well-discussed, it would be beneficial to have a comparative discussion with competing methods, highlighting where BINDER outperforms them in more depth. This would provide additional context for readers.

5) The complexity. It seems that the model requires very high dimensionalty but there is no such discussion. The worst case is that the concepts are fully disjoint, then you need N dimension, which makes the model not scalable.

**Questions:**

1) Why does the accuracy (acc) decrease when the number of transitive edges increases? How can this phenomenon be explained?

2) In equation (1), you specify that a $\neq$ b, meaning when a = b, it should be considered a negative example. However, in Section 2.3, the negative pairs (n) involving reflexivity constraints, where a is-a a, are excluded. Does this mean that reflexivity constraints are neither treated as positive nor negative examples? I understand that in the experiments and method, you've avoided dealing with the self-relation, but in practice, it might occur. How do you plan to handle cases where the reflexivity constraint is present?

3) In Section 2.3 (Training Algorithm), it might enhance clarity and conciseness by describing the sampling process as selecting $r$ from $W \setminus \{a, b\}$. This would align well with the earlier statement that the number of negative examples is $n^2 - n - |P|$.

4) In Table 3 (Reconstruction Results Acc(\%) (dim)) and similar cases, what is the purpose of including "(dim)" in the table header?

5) One notable advantage of BINDER is its ability to generate embeddings for unseen concepts using logical functions over existing concepts, a feature not present in its competitors.

6) Could the finite permutation space of binary vectors result in a loss of expressivity, limiting the model's ability to capture complex relationships?

---

> ### Author Response · Authors · 2023-11-20
> **Response to Reviewer EAos**
>
> Thank you very much for your insightful review.  Below is our response:
>
> Weakness comments:
>
> W1: Generalization to Unseen Concepts: While BINDER claims to generate embeddings for unseen concepts by using logical functions over existing concepts, it would be beneficial to provide more detailed explanations and examples of how this generalization is achieved.
>
> A: All the existing works that we compare against, along with BINDER follows a transductive learning setup, so they cannot generate embedding of arbitrary unseen objects. This is the reason, in our paper and in earlier works, link prediction is evaluated over transitive closure edges, for which the entities of the predicted edges are already present in the training data. However, BINDER being binary
> vector based representation, is able to provide embedding of some entities, which are direct logical derivative of existing entities in the
> training data. One example that we gave in the paper is, if BINDER generate an embedding of "living-thing", and "non-living-thing" is not in the training data, we can simply inverse the bits of "living-thing" to give an embedding of non-living thing. In general, if an entity
> $X$ can be logically derived by a boolean formula over entities in the training data, Say, $X = formula(S)$, where $S$ is a subset of entities in the training data, BINDER can obtain the embedding of $X$ as following: $Emb(X) = formula(Emb(S))$. This is how, BINDER can obtain limited generalization. Although this is not full generalization, it is better than the competitors method, which provide no generalization at all.
>
> W2: Experimental Rigor: The paper mentions that BINDER is a randomized algorithm but provides results from the best run out of five. It would be helpful to include more detailed information on the variability observed in these runs, such as mean and standard deviation. A discussion of the algorithm's sensitivity to random initialization would also be insightful.
>
> A: In Table 5 we have provided mean and standard deviation for BINDER’s experimental results.
>
> W3: Hyperparameter Sensitivity: The paper discusses hyperparameters like the learning rate and bias but does not delve into their sensitivity analysis. A study on how these hyperparameters affect BINDER's performance and convergence would provide a better understanding of its behavior.
>
> A: In appendix C we have discussed about hyperparameter tuning and listed the optimum hyperparameters for our experiments. However, we agree with the reviewer that hyperparameter sensitivity experiment would provide a better understanding of BINDER's behavior. If the paper is accepted, we will add another result section in Appendix, discussing "hyperparameter sensitivity".
>
> W4: Comparative Discussion: While BINDER's strengths are well-discussed, it would be beneficial to have a comparative discussion with competing methods, highlighting where BINDER outperforms them in more depth. This would provide additional context for readers.
>
> A: BINDER outperforms competing models particularly in transitive closure experiments. BINDER only needs direct edges to learn all entity embeddings in the training set. This is validated in the  0% transitive closure experiments; in these experiments, BINDER can easily achieve close to 99% accuracy, whereas the competing models perform poorly, with accuracy in the range of 60% to 90%. We will update our manuscript to highlight BINDER's strength.
>
> W5: The complexity. It seems that the model requires very high dimensionality but there is no such discussion. The worst case is that the concepts are fully disjoint, then you need N dimension, which makes the model not scalable.
>
> A: Since binary space has only two possible values at each dimension, it is expected that binary embedding will need many bits to capture the desired is-a relationships, specifically for large datasets. However, do note that, BINDER works on bit domain, whereas existing SOTA works on real-vector domain. For BINDER, a dimension of 200 is simply 200/8 = 25 bytes, which is space taken by  6 real numbers.
>
> Questions:
>
> Q1: Why does the accuracy (acc) decrease when the number of transitive edges increases? How can this phenomenon be explained?
>
> A:  General trend is that when the transitive edges in the training data increase, the performance of the learning task increase. This is indeed the case in Table 4 as we gradually increase Transitive closure from 0% to 50%. There is exactly 3 exceptions in Table 4. Those
> are:  (1) Medical dataset (10% →  25%), (2) Music (0% →  10%) and Nouns (25% →  50%); in these three cases there are slight drops in accuracy, which can probably be attributed to  BINDER's randomized algorithm for learning embedding.

---

> > ### Author Response · Authors · 2023-11-20
> > **Response to Reviewer EAos (continuing)**
> >
> > Q2: In equation (1), you specify that a $\ne$ b, meaning when a = b, it should be considered a negative example. However, in Section 2.3, the negative pairs (n) involving reflexivity constraints, where a is-a a, are excluded. Does this mean that reflexivity constraints are neither treated as positive nor negative examples? I understand that in the experiments and method, you've avoided dealing with the self-relation, but in practice, it might occur. How do you plan to handle cases where the reflexivity constraint is present?
> >
> > A:  If we allow $a=b$, then an all-zero vector for all entities will satisfy all constraints $a \cap b = b, \forall (a, b) \in P$, trivially. $a \ne b$ is needed solely to avoid that. By explicitly ensuring that $a \ne b$, we guaranty that every entity has a distinct bit-vector. Yes, you are right, reflexivity constraints are neither treated as positive nor negative examples. In practice, if $a = b$ occurs, we will first pre-process the dataset to cluster all equal entities to create super-entities; this will be like contraction of the edge $(a, b)$ (as is done in Karger's min-cut algorithm). After such contraction, all remaining entities are distinct and BINDER can work as usual.
> >
> > Q2: In Table 3 (Reconstruction Results Acc(%) (dim)) and similar cases, what is the purpose of including "(dim)" in the table header?
> >
> > A: Since, BINDER uses bit-vector, it provides massive saving in terms of memory footprint of the embedding vectors. To compare BINDER's memory footprint with other competing methods, we were really interested to see at what dimension other competing methods reach their best results. As we have harvested those results, we thought to include this in this Table
> > for the readers to see that not all methods reach their best for similar dimension value. This could instigate a follow-up work
> > by us or others to investigate how the dimension value affects the performance of different embedding models, which may
> > use different kind of geometric space. For instance, a general observation from this results is that HEC (Hyperbolic Entailment Cone) embedding generally requires higher dimension value to achieve best result.
> >
> > Q3: One notable advantage of BINDER is its ability to generate embeddings for unseen concepts using logical functions over existing concepts, a feature not present in its competitors.
> >
> > A: We take this as a comment, not as a question and refer you to our answer to your W1 comment.
> >
> > Q: Could the finite permutation space of binary vectors result in a loss of expressivity, limiting the model's ability to capture complex relationships?
> >
> > A:  It is possible, and that is why a randomized algorithm is the best to obtain a solution for such an embedding framework, like ours. We believe by using randomization, we are able to exploit the best out of a finite permutation space of binary vectors. More research is needed to understand BINDER's limit for capturing complex relationship. In this work we confined ourself only to is-a relationship, which is transitive, and easy to work with (non-complex). But, we will claim that BINDER can be generalized to extend to other relationships, which are more complex and in that scenario BINDER's ability can be put into test.
> >
> > Thank you very much for your time to review our paper.

---

> > > ### Comment · Reviewer_VtjX · 2023-11-22
> > > **Thank you for your rebuttal**
> > >
> > > Thank you for the detailed rebuttal, I have read them in detail, and respond to them below.
> > >
> > > Q1: There is a simpler deterministic solution to your stated task (which also uses bit-vectors), for which it would obtain a perfect score on your 0% transitive closure test. Why is this not a preferable approach? Based on your other replies, I think you would claim that the problem is that the deterministic algorithm would not generalize, to which I would say (a) your current evaluation is too weak to claim it supports generalization (more on this later), and (b) the "generalization" you claim is not well-defined, in the sense that for any graph where you would exhibit "generalization" there is another graph which is equally likely given the training data for which you would not "generalize". This problem is simply not well-defined. In addition, the extent to which you can claim superiority based on empirical evaluation depends on both ensuring adequate baseline coverage and the rigor of the evaluation, both of which I take issue with (and address in subsequent questions).
> > >
> > > Q2: I agree that SAL does not provide a fixed-size vector embedding and does not capture order between entities, whereas BINDER does. I do not agree that you have demonstrated BINDER's representations transfer to other subsequent knowledge discovery tasks. As for the calculation regarding space efficiency, this assumes BINDER's bit vectors perfectly capture the adjacency matrix, which has not actually be evaluated. It is possible that with 100 dimensional bit vectors you cannot perfectly capture the adjacency matrix. (In fact, the deterministic algorithm should allow for a constructive proof which also allows you to determine a sufficient dimensionality, and perhaps even the minimum dimensionality under which your proposed scheme would capture an adjacency matrix.) To make this even clearer, the current eval dataset only checks BINDER on a very limited subset of the entities in the adjacency matrix, for which it would be straightforward to design an adjacency-list method which obtains perfect accuracy on this subset with far fewer bits than BINDER.
> > >
> > > Q3: What is the practical benefit of using the transitive reduction and "generalizing" to the transitive closure? If one knows the relation is transitive, this isn't really generalization, you may as well just have learned the original graph and applied a transitive closure operation as a post-hoc procedure, which is easy to implement and would also obtain perfect performance on the proposed evaluation.
> > >
> > > Q4: You state "one can extrapolate embedding of entities which are not in the training data by using logical operations over BINDER's embedding vectors", and you contrast this with cone embeddings where the "inverse" cone of (0.5, 0.5, ..., 0.5) is itself, however this is not imposed by the structure of the cone embeddings. The *complement* of a cone is the whole space set-difference with the cone. It is not, itself, representable as a cone, that is true, however it is well-defined and does exist and, moreover, appropriately captures the set-theoretic notion of a complement. On the other hand, your proposed definition is, indeed, a way to assign a bit-vector to the "inverse" of an existing embedding, but as you agree with my example of how this applies to "living thing" it appears this "inverse" is not actually useful, in so far as your proposed inverse of "living thing" (a) does not have all non-living things as children, (b) there are elements which are neither living or non-living, and (c) there are elements which are both living and non-living. One could similarly define such operations for other embeddings which are not self-consistent.
> > >
> > > Q5: It is not clear to me that your randomized algorithm offers any generalization either, because the evaluation is too weak. Moreover, in a very clear sense, any proposed algorithm which "generalizes" in the sense that you describe must also necessarily generalize *incorrectly* for other graphs. To put this another way: if your only assumption is "the following edges are taken from a transitively-closed directed graph" and the goal is to select some transitively-closed directed graph containing those edges, then a priori each of them are equally likely, and therefore no generalization is possible. If, on the other hand, you are claiming that your particular style of randomized algorithm somehow has a useful (but unspecified) inductive bias toward *real-world* "is-a" hierarchies, I think this is entirely unsupported by the work.
> > >
> > > I think my earlier replies also raises concerns with your responses in Q6-Q8. I will say, however, that the evaluation on BLESS is not particularly enlightening, since it was not compared with any baseline. To the extent to which comparisons can be made, however, other baselines have obtained higher scores on larger graphs (see [0] from my original reply).
> > >
> > > I appreciate your replies, however all concerns from my original review remain.

---

> > > > ### Author Response · Authors · 2023-11-23
> > > > **Response to Reviewer VtjX**
> > > >
> > > > Thanks for your further response. We appreciate it!
> > > >
> > > > We have considered Animals dataset for evaluating the representational capacity of the full adjacency  matrix as you suggested in your original response. From our experiment we got the following results:
> > > >
> > > > BINDER (Balanced Accuracy): 99.82 (maximum), 99.4 ± 0.5 (mean ± standard deviation).

---

### Official Review · Reviewer_JqKj · 2023-10-31

**Soundness:** 2 fair
**Presentation:** 2 fair
**Contribution:** 3 good
**Rating:** 3
**Confidence:** 4

**Summary:**

This work proposes BINDER, an approach for order-based representation. BINDER uses binary bits as representation vectors, via a scalable optimization procedure. Authors evaluate experiments on both prediction and reconstruction tasks.

**Strengths:**

Overall, the paper is well-organized, and the authors provide a detailed description of their contributions.

**Weaknesses:**

1. The Introduction section is also missing an important recent work on two-view knowledge graph embeddings, which jointly embed both the ontological and instance view spaces:
[KDD 2022] Dual-Geometric Space Embedding Model for Two-View Knowledge Graphs. In Proceedings of the 28th ACM SIGKDD Conference on Knowledge Discovery and Data Mining (KDD '22). Association for Computing Machinery, New York, NY, USA, 676–686. https://doi.org/10.1145/3534678.3539350

2. It would be helpful if the authors could create an illustration of an example knowledge graph following their problem formulation.

3. Further, the model fails to include important baseline models such as standard knowledge graph embedding model in the hyperbolic space e.g., RefH/RotH/AttH, hyperbolic GCN (HGCN), and the product space (M2GNN).

4. Moreover, the size of the datasets also seem to be relatively small-scale with number of nodes and edges on the scale of thousands as opposed to million node/billion edge graphs indicative of real world KGs e.g., DBPedia & YAGO.

**Questions:**

Why is only the hyperbolic space being considered? Entities can form cyclic relations as well, which is better modeled in the spherical space.  Perhaps the authors need to more clearly denote the distinction between entities and concepts.

---

> ### Author Response · Authors · 2023-11-19
> **Response to Reviwer JqKj**
>
> We appreciate Reviewer's time to review this paper. Below is our responses:
>
> Response to Weakness comments:
>
> **Comment 1:** The Introduction section is also missing an important recent work on two-view knowledge graph embeddings, which jointly embed both the ontological and instance view spaces: [KDD 2022] Dual-Geometric Space Embedding Model for Two-View Knowledge Graphs. In KDD '22
>
> **Answer:** Our work is not a general knowledge graph embedding paper, so it is not clear why the reviewer is considering "two-view knowledge graph embedding" an important recent work related to our contribution. In our entire manuscript (except the references) the phrase "knowledge graph" came only twice (both in second paragraph of introduction); just to clarify that we are concerned with only hierarchical concept representation, not for embedding of RDF like knowledge graph entities. So, it is not clear to us in what context we will cite or discuss the paper that the reviewer is referring to. We would appreciate if the reviewer kindly clarify this to us.
>
> Just to reiterate: given a collection of entities where the entities are related by is-a relation, we are designing order embedding representation of these entities so that the entity vectors can capture the order relation. This is our task, which is also the task of the works that we are comparing against in our experiment section. The KDD 2022 paper is neither related to us, nor can we use the dataset used in that paper (Yago, DbPedia) to compare against.
>
> **Comment 2:** It would be helpful if the authors could create an illustration of an example knowledge graph following their problem formulation.
>
> **Answer:** Continuing from our response from Comment 1, it is not clear to us why would we create an example knowledge graph, when our problem definition and solution does not pertain to a knowledge graph embedding task. However, we do have an example graph in Figure 1 (See the discussion of Appendix D) to show is-a relations among a few entities. It is actually a tree-like graph as is-a relation is transitive.
>
> **Comment 3:** Further, the model fails to include important baseline models such as standard knowledge graph embedding model in the hyperbolic space e.g., RefH/RotH/AttH, hyperbolic GCN (HGCN), and the product space (M2GNN).
>
> **Answer:** This work is not a model for "knowledge graph embedding in the hyperbolic space". Could the reviewer kindly explain why and how the listed works are important baselines?
>
> **Comment 4:** Moreover, the size of the datasets also seem to be relatively small-scale with number of nodes and edges on the scale of thousands as opposed to million node/billion edge graphs indicative of real world KGs e.g., DBPedia & YAGO.
>
> **Answer:** We used the largest labeled datasets that is publicly available (WordNet Nouns) for the order-embedding. Earlier works also use this dataset. In fact, several of the earlier works use only one dataset, we are using 5 datasets to produce a comprehensive result.
>
> Response to Questions:
>
> **Question 1:** Why is only the hyperbolic space being considered? Entities can form cyclic relations as well, which is better modeled in the spherical space. Perhaps the authors need to more clearly denote the distinction between entities and concepts.
>
> **Answer:** Hyperbolic space is being considered because hyperbolic cones have been used in earlier works for performing order embedding of entities exhibiting is-a relationship. Is-a relation is by definition non-cyclic, so cyclic relations do not occur in our dataset and our problem formulation. Spherical space is not ordered space, nor is it used for ordered embedding, so it is not considered.
>
> We are using entity and concept synonymously in this paper, as both are relevant when someone refer to hypernym-hyponym (is-a) relationship. For example, we can think of "classification model" as a concept in Machine Learning, a sub-concept of this can be "Neural Networks"; Neural Networks is-a Classification model, so they exhibits an ordered is-a relation, which is clearly not cyclic. On the other hand "cat" is an entity, and "animal" is another entity. Cat is-a animal, and they exhibit an ordered is-a relation, which is also non cyclic.
>
> **We suspect that the reviewer completely misunderstood the context of our work. We request the reviewer to kindly read the paper once again, after reading our responses. Our suspicion comes from the fact that except the first two sentences of the review, no other comments in this review appear to be relevant to our paper and our contribution.**

---

> > ### Comment · Reviewer_JqKj · 2023-11-23
> >
> > Thanks to the authors for providing a response to the questions raised. However, overall, your answers to the above questions still leave us with significant concerns, so I cannot raise my score. However, it seems that the paper is promising and with appropriate revision has good potential in another future venue.

---

### Official Review · Reviewer_VtjX · 2023-11-03

**Soundness:** 3 good
**Presentation:** 3 good
**Contribution:** 1 poor
**Rating:** 1
**Confidence:** 5

**Summary:**

In this work the authors develop a method for creating bit-vector representations of entities such that an order relation on bit-vectors captures some hierarchical structure. More specifically, the authors focus on representing hypernym ($\texttt{is-a}$) relationships between entities. For a given set of entities $W$ and some set of $\texttt{is-a}$ relationships expressed as pairs $P \subseteq W \times W$, the authors propose to represent each entity $a \in W$ by a bit-vector $\mathbf a \in \\{0,1\\}^d$ for some $d$, such that

$$\mathbf b_j = 1 \implies \mathbf a_j = 1 \quad \iff \quad (a,b) \in P \setminus \Delta_W,$$

where $\Delta_W = \\{(a,a) \mid a \in W\\}$ is the identity relation on $W$.

The authors formulate a loss function for a bit-vector representation of entities which is a linear combination of a "positive loss", which counts the number of times $(a,b) \in P$ and $a\ne b$, but there is some $j$ for which $\mathbf b_j = 1$ and $\mathbf a_j = 0$, and a "negative loss", which counts the number of times $(a,b) \in N \subseteq W \times W \setminus (P \cup \Delta_W)$ are such that $\mathbf b_j = 1 \implies \mathbf a_j = 1$. Since the representation is discrete we cannot take gradients of this loss function, so the authors propose an algorithm which randomly flips bits with a probability which is correlated with the amount of improvement in the loss function as a consequence of flipping that bit.

They evaluate their model on 5 hypernym datasets. They evaluate in both a reconstruction setting as well as a setting where edges from the transitive closure are removed during training and expected to be recovered during evaluation. They claim their model generally outperforms  baselines including order embeddings, Poincare embeddings, and hyperbolic entailment cones.

**Strengths:**

The authors do an admirable job presenting the background and motivation for this work. Their proposed model is explained clearly, and the randomized algorithm they propose is somewhat novel.

**Weaknesses:**

Unfortunately, there a many fundamental problems with this work.

First, it is unclear to me what problem or task the proposed model is actually solving. What do we gain by representing entities with bit vectors capturing their hypernym relationships? In general, the motivation to embed entities in this setting is one of the following:
1. Space Efficiency: The new representation requires fewer bits to store than some naive approach (eg. adjacency list of the transitive reduction)
2. Computational Efficiency: There is some operation which can be performed on the embedded representation more efficiently than on some other representation
3. Generalization: The embedding allows one to infer missing edges between existing nodes or make predictions of graph edges from unseen nodes (based on input node features)
4. Transference to Other Tasks: The embedding captures the graph relationships which can then be plugged into other architectures for use in tasks which benefit from the knowledge of the graph structure (eg. MLP for classification)

The authors discuss space efficiency in Appendix F.3, however comparisons here are only made to other baselines, and the numbers quoted are far and above what would be required (eg. the authors claim that baselines with more than 100 dimensions take more than 10 hours to run, but this is far longer than the numbers reported in [0] and my personal experience suggests, where it is possible to train a model to represent WordNet reasonably well in 10-20 minutes). Comparing bit vectors to floating point models which were not quantized is disingenuous at best. The authors do claim their embedding is useful for is generalization, however the evaluation performed only assesses generalization to the transitive closure, which is trivial to perform symbolically on the set $P$ which would result in perfect accuracy on this evaluation. There are also issues with this evaluation separately, which are addressed below, but fundamentally this task is not truly a test of generalization in any useful sense.

The authors do claim that Binder embeddings have some unique capabilities unavailable to other models. Specifically, they claim that Binder embeddings have a well-defined complement, union, or intersection, however this is not true, or at least not any more true here than in any other embedding method. The authors even state that if "we have a concept 'living-thing' for which we have a binary vector representation, [...] if we want to obtain a representation for 'not living things' we can obtain that simply by reversing the bits of the 'living thing' vector', however this is not true. To see this, consider a "living thing" vector as $[0,1,1]$, then based on the authors' embedding definition the set of living things is $\{[0,1,1],[1,1,1]\}$. By their claim, the representation of "not living thing" should therefore be $[1,0,0]$. This would mean that the set of living things includes the bit vectors $\\{[0,1,1],[1,1,1]\\}$ and the set of not living things is $\\{[1,0,0],[1,1,0],[1,0,1],[1,1,1]\\}$. Note that this means that the bit vector $[1,1,1]$ is both living and not living. Moreover, it also means the space is not decomposed into just "living thing" and "not living thing" - for example, the vector $[0,1,0]$ is neither living or not living. Therefore this definition of complement is not correct. Not only that, there is *no* bit vector which captures the full complement of being a living thing, because to not be a living thing, according to their definition, we simply need to have a zero in the first or second position, and there is no way to express this "or" condition with a single vector. A similar argument shows Binder embeddings are not closed under union.

Secondly, even if there is some benefit to representing entities by bit-vectors, it is straightforward to provide a deterministic algorithm which takes a set $P$ and produces a bit-vector embedding which perfectly satisfies the constraint above using a topological sort. With some additional care in the construction process, it even seems possible to create a bit-vector with minimal size which perfectly satisfies the constraint. Therefore, the use of a randomized algorithm here does not seem to have any benefit.

Thirdly, there are a number of problems with the experiments. For some reason, the authors chose to report a reweighted accuracy statistic as opposed to the more conventional F1 metric when dealing with data imbalances. In addition, the authors evaluate on a test set with negatives which were created by random perturbation, however this approach can lead to a very coarse evaluation, and has issues with test set bias. For the test set accompanying Order Embeddings paper, for example, you can get almost 0.90 F1 by simply treating any node in the training data which has a child as though it is a parent to every other node in the training set. It was for this reason that more comprehensive evaluations advocate for using the full adjacency matrix [0]. In addition, the other models present in that paper all serve as reasonable baselines, and the [associated code](https://github.com/iesl/geometric-graph-embedding) has implementations readily available.

Finally, a number of the characterizations or claims made in the introduction are incorrect. The authors claim optimization algorithms are not well studied for hyperbolic space, however this is not the case - Riemannian gradient descent is well understood ([1], [2], [3]). Moreover, there are approaches to parameterizing and training on hyperbolic space which have been shown empirically to work well with standard gradient descent techniques such as SGD or Adam [4]. The authors claim box embeddings have more degrees of freedom than point embeddings, but this is not true - a box embedding in $d$-dimensional space does have $2d$ parameters per box, but it is for this reason that experiments using box embeddings compare $d$-dimensional boxes to $2d$-dimensional vectors, so they have exactly the same number of free parameters. The claim that bit vectors are more interpretable is not supported by any experiments, and there is no clear reason to expect that the randomized algorithm leads to interpretable properties in each dimension. The interpretability hinted at for the bit vectors is equivalent to the level of interpretability that order, probabilistic order, or box embeddings provide.

[0] Boratko, Michael, et al. "Capacity and bias of learned geometric embeddings for directed graphs." Advances in Neural Information Processing Systems 34 (2021): 16423-16436.
[1] Bonnabel, Silvere. "Stochastic gradient descent on Riemannian manifolds." IEEE Transactions on Automatic Control 58.9 (2013): 2217-2229.
[2] Bécigneul, Gary, and Octavian-Eugen Ganea. "Riemannian adaptive optimization methods." arXiv preprint arXiv:1810.00760 (2018).
[3] Hu, Jiang, et al. "A brief introduction to manifold optimization." Journal of the Operations Research Society of China 8 (2020): 199-248.
[4] Law, Marc, et al. "Lorentzian distance learning for hyperbolic representations." International Conference on Machine Learning. PMLR, 2019.

**Questions:**

The section on the weaknesses highlights my concerns with this work.

1. Can you clarify the specific problem or task that your proposed model is designed to solve? How does the use of bit vectors capturing hypernym relationships contribute to solving this problem?

2. Regarding space efficiency, how does the bit vector representation compare to a sparse adjacency list? If it is not more compact, does it offer any benefits beyond the sparse adjacency list?

3. In terms of generalization, the current evaluation focus on the transitive closure. When training on this data in the 0% case, does your negative set include edges from the transitive closure? Regardless, if we know the relation is transitive, what benefit do we gain by training on the transitive reduction and being able to "generalize" to the transitive closure which is not also achievable by simply taking the transitive closure of the training data?

4. You mention unique capabilities of Binder embeddings, such as well-defined complement, union, or intersection operations. Given the issues highlighted with these operations, how do you respond to the concerns about the correctness of these claims?

5. Could you elaborate on why a randomized algorithm is used for generating bit-vector embeddings when a deterministic algorithm could suffice?

6. Why was a reweighted accuracy statistic chosen over the conventional F1 metric in your experiments, especially in the context of data imbalances?

7. Please correct or respond to my assertions above regarding the inaccuracies in characterizing other baselines. (For example, the assertion that optimization algorithms in hyperbolic space are not well-studied,  or that box embeddings have more degrees of freedom than point embeddings.) After correcting these claims, please address what specific benefits this embedding provides beyond those provided by the baselines.

8. The interpretability of bit vectors is claimed to be superior in your paper. Can you provide empirical evidence or a theoretical framework that supports this claim, in a setting where equivalent effort is also given to order, probabilistic order, or box embeddings?

9. Would you consider evaluating the representational capacity of your model on the full adjacency matrix? This may be computationally prohibitive; if so, it is reasonable to select a subgraph (eg. Animal subgraph from WordNet) and evaluate the full adjacency matrix on that subgraph.

---

I apologize if my review seems harsh. I would like to commend the authors for the clear and structured presentation of their approach. The manuscript is well-written, and the methodology is articulated with a level of detail that reflects a thorough understanding of the subject matter. It is evident that considerable effort has gone into developing and describing the proposed model.

One of the main challenges in this area of research seems to be a legacy of ambiguity in motivations and intentions from previous works, like a bad game of "telephone". Previous evaluations designed to highlight specific aspects of a model may be misconstrued to be a task in and of themselves, and this problem can compound on itself in subsequent work. It is also possible that I misunderstood the author's motivations and approach, and if so then I humbly apologize and ask the authors to clarify things for me.

---

> ### Author Response · Authors · 2023-11-19
> **Response to Reviewer VtjX**
>
> We found "strong reject" to be unfairly harsh; nevertheless, we sincerely appreciate your detailed review of our paper. Below we provide counter arguments to your criticisms and will appeal to you to change your rating in our favor.
>
> Answer to Questions:
>
> Q1: Can you clarify the specific problem or task that your proposed model is designed to solve? How does the use of bit vectors capturing hypernym relationships contribute to solving this problem?
>
> A: We solve the following task: *From a collection of is-a relations between a pair of entities, obtain embedding of the entities such that embedding vectors capture the order imposed through the is-a relation*. Use of bit vectors as representation vectors as obtained from BINDER algorithm captures the imposed order efficiently, and effectively, thereby solving the problem. Not only BINDER solves the problem, but it does so better than existing state-of-the-art, which we established through comprehensive experiments: we show that BINDER performs better on reconstruction and transitive closure link prediction than competing models as data size gets larger. Our problem formulation and experimental setup is identical to competing methods; but we have better results than those methods.
>
> The contribution of this work is innovative, and novel, both in terms of embedding idea and optimization framework. In our experiments, we have shown BINDER is better in terms of **space efficiency** and **computational efficiency** (see Appx. F.3). BINDER is also better in **generalization** in transductive setup as per 0% transitive closure results (note: for this task, all existing SOTA methods are transductive). BINDER's binary embedding can be plugged into other architectures (a NN can take a binary vector as input) to use in subsequent knowledge discovery tasks.
>
> Q2: Regarding space efficiency, how does the bit vector representation compare to a sparse adjacency list? If it is not more compact, does it offer any benefits beyond the sparse adjacency list?
>
> A: Sparse adjacency list (SAL) does not provide fixed-size vector embedding of entities, but BINDER's bit-vector representation provides that. SAL does not capture order between entities, but BINDER's bit-vector provides that. SAL only captures the edges of the relation, but BINDER's bit-vector is an order-embedding, which represents nodes as vectors which are transferable to other subsequent knowledge discovery task.
> For the Noun dataset, which has 743K edges and 82K vertices, BINDER's bit vector will take 82K * 100/8 $\approx$ 1025 Kbytes, whereas a sparse adjacency list will take at least (743K+82K) * 4 = 3300 Kbytes (considering 4 bytes for integer).
>
> Q3: In terms of generalization, the current evaluation focus on the transitive closure. When training on this data in the 0% case, does your negative set include edges from the transitive closure? Regardless, if we know the relation is transitive, what benefit do we gain by training on the transitive reduction and being able to "generalize" to the transitive closure which is not also achievable by simply taking the transitive closure of the training data?
>
> A: Our training data for the 0% transitive closure link prediction experiment uses only the direct edges as the positive training set. During training, "negative" samples are generated entirely at random, so they could randomly be actually in the transitive closure.
> However, such chances are extremely small due to sparsity of the edges in the dataset. In order embedding, to show generalization, one must establish that the algorithm can assign a vector to entities from which unseen order between entities can be inferred; so the benefit of training using minimal set of edges is that one can then prove the model's "capability to ordering entities" by showing its prediction over the unseen transitive edges.
>
> Q5: Could you elaborate on why a randomized algorithm is used for generating bit-vector embeddings when a deterministic algorithm could suffice?
>
> A: A deterministic algorithm offers no learning, it is simply memorizing the edges. It can only be used , when entire minimal set of edges of the DAG is given. But, one cannot expect that all the edges between entity pairs are already known/given in the training data. if that is the case, no learning or embedding is needed and a fixed deterministic method can be used.
>
> The advantage of BINDER is that it has learning capability. It assign bit-vectors so that it can infer missing edges. In other words, if we remove some direct edges in the training data, BINDER will still be able to embed entities reasonably. We have performed Link Prediction experiments to prove this claim on Mammals and Nouns dataset. For Mammals dataset (edges=6.5k) and Nouns dataset (edges=743k) we randomly take out 300 and 3500 edges respectively to construct test data. Here is the result on this test data.
> Mammals: Acc (94.2 ± 1.1) , F1 (94.1 ± 1.1);   Nouns: Acc (93.9 ± 0.4), F1( 93.6 ± 0.5)

---

> ### Author Response · Authors · 2023-11-20
> **Response to Reviewer VtjX (continuing)**
>
> Q4: You mention unique capabilities of Binder embeddings, such as well-defined complement, union, or intersection operations. Given the issues highlighted with these operations, how do you respond to the concerns about this?
>
> A: Complement, union, intersection are well-defined for binary bits, so using BINDER embeddings, one can extrapolate embedding of entities which are not in the training data by using logical operations over BINDER's embedding vectors. Our claim extends only up to this. We are not claiming that BINDER's embedding will ensure logical inference and is-a relation consistency over entity-pairs, which are not in the training data. We claimed that other continuous domain models do not have well defined complements. For example, in a continuous domain, the “inverse” cone of (0.5, 0.5, ..., 0.5) is itself, which makes little sense when such a vector represents a concept in a domain. The scope of our work is to highlight the benefits of binary vector embedding to capture order relationships over the competing models in terms of simplicity of the optimization algorithm, compactness of the embeddings, time-efficiency, and generalization over unseen entity up to certain capacity.
>
> Considering reviewer's example, say, based on BINDER, living-thing has a representation of $[0, 1, 1]$. Then based on our claim, non-living thing has an embedding of $[1, 0, 0]$. Our claim extends only up to this. Any transitivity and is-a extension over the non-living entities is out of scope of BINDER as the method is transductive and has no capability to ensure consistency over relations between unseen pairs (again, all SOTA methods on this task are transductive). So, when the reviewer
> extends is-a relation over the set of not living and suggests that the set of non-living things are: $\{[1, 0, 0], [1, 1, 0], [1, 0, 1], [1, 1, 1]\}$, BINDER does not endorse that claim.
>
> We are sorry if we over-claimed in our write-up which created the confusion. We will clarify our claim in revised manuscript.
>
> Q6: Why was a re-weighted accuracy statistic chosen over the conventional F1 metric in your experiments, especially in the context of data imbalances?
>
> A: We agree that for classification over imbalance data, F1 measure is a good metric; however some argue (and we agree with them) that balanced_accuracy (BA) is actually a better metric, which is defined as the average of positive class recall = $\frac{TP}{TP+FN}$, and negative class recall=$\frac{TN}{TN+FP}$. Balanced accuracy is also defined as $\frac{sensitivity + specificity}{2}$. Balanced accuracy is a better metric than F1-score for imbalanced dataset, because F1-score does not care about how many true negatives are being classified. This is because F1-score uses precision and recall which together use only three entries of the confusion matrix (TP, FP, FN); on the other hand, balanced accuracy uses all four entries of the confusion matrix. For an example, say a dataset has 1000 (-ve) and 10 (+ve) examples. If the model predicts there are 15 positive (TP=5, FP = 10), and predicts the rest as negative (TN = 990, FN = 5), we get, $Precision=\frac{5}{15}= 0.33$, and $Recall = \frac{5}{10} = 0.5$, it yields, $F1-score = 0.4$. The model gets absolutely not much credit for correctly predicting 990 out of 1000 examples as negative. However, the balanced accuracy $= \frac{1}{2} (\frac{5}{10} + \frac{990}{1000}) = 0.745$, which provides somewhat a more realistic picture.
>
> We actually used F1-measure, but found it to be very harsh for the competitors. We want to be nicer to the competitors and used Balanced accuracy. In the Table below, we show Precision, Recall, and F1-score (left to right) results for 0% transitive link prediction. As can be seen BINDER's F1-Scores (bold numbers) are significantly better than all other methods on all datasets. The competitor methods suffers severely due their poor precision.
>
> | Models   | Medical                | Music                  | Shwartz Lex            | Shwartz Random         | Nouns                  |
> | -------- | ---------------------- | ---------------------- | ---------------------- | ---------------------- | ---------------------- |
> | OE       | 70.1, 97.6, 81.6       | 43.8, 94.6, 59.9  | 33.1, 79.3, 46.7           | 23.6, 83.4, 36.7       | 49.6, 48.2, 48.9     |
> | Poincare | 35.2, 95.2, 51.4 | 19.6, 66.2, 30.3 | 21.1, 80.6, 33.5 | 24.4, 59, 34.5  | 40.1, 28.1, 33.1 |
> | HEC      | 44.9, 83.3, 58.3     | 27.7, 61.5, 38.2    | 23.3, 80.6, 36.1        | 31.1, 61, 41.2          | 43.7, 36.2, 39.6  |
> | T-Box    | 18, 88, 29.8       | 17.5, 93.1, 29.5    | 14.8, 94.2, 25.5         | 13.2, 83.2, 22.8          | 14.8, 97.8, 25.7 |
> | BINDER   | 93.3, 100.0, **96.6** | 78.3, 100.0, **87.8**  | 96.8, 100.0, **98.4** | 95.1, 100.0, **97.5**  | 85.2, 99.2, **91.7**  |

---

> ### Author Response · Authors · 2023-11-20
> **Response to Reviewer VtjX (continuing)**
>
> Q7: Please correct or respond to my assertions above regarding the inaccuracies in characterizing other baselines. (For example, the assertion that optimization algorithms in hyperbolic space are not well-studied, or that box embeddings have more degrees of freedom than point embeddings.) After correcting these claims, please address what specific benefits this embedding provides beyond those provided by the baselines.
>
> A: We will correct the claims in the revised manuscript, as you suggested. However, after correcting these claims, BINDER still holds the following specific benefits over competitors: (1)  better performance, (2) lower memory footprint of the algorithm, and (3) Ability to get embedding of out-of-training data in a limited setting (through binary operation over bit vectors).
>
> Q8: The interpretability of bit vectors is claimed to be superior in your paper. Can you provide empirical evidence or a theoretical framework that supports this claim, in a setting where equivalent effort is also given to order, probabilistic order, or box embeddings?
>
> A: BINDER is inspired from Formal Concept Analysis (FCA), which uses intent-extent ideas to represent relationships between a concept and subconcept. We tried to draw a parallel with FCA and alluded that '1' entries in a BINDER embedding corresponding to "presence of a latent property". This gives us a relative idea of how many latent properties are shared by two concepts and how strong is their relationships. We want to refer to Appendix D where we show a case study experiment on a toy dataset. At present, we do not have any theoretical framework to support this claim. This is a hypothesis only, and we will address BINDER embedding explainability in future works.
>
> Q9: Would you consider evaluating the representational capacity of your model on the full adjacency matrix? This may be computationally prohibitive; if so, it is reasonable to select a subgraph (eg. Animal subgraph from WordNet) and evaluate the full adjacency matrix on that subgraph.
>
> A: As you understand, this is computationally prohibitive. For example, the complement of Noun dataset would involve more than 6 billions of negative edges. So, we have considered BLESS dataset (Link: https://github.com/alexanderpanchenko/sim-eval/blob/master/datasets/bless.csv), which has 330 entities. We considered 0% transitive closure experiment; In this setup train data had all the 1300 direct edges. Test dataset has indirect positive edges and the negative edges consist of the remaining edges of the full adjacency matrix. From our experiment we got the following results:
>
> BINDER (Balanced Accuracy): 99.2  (maximum), 98.4 ± 0.8 (mean $\pm$ standard deviation).

---

### Official Review · Reviewer_psGs · 2023-11-05

**Soundness:** 2 fair
**Presentation:** 3 good
**Contribution:** 2 fair
**Rating:** 3
**Confidence:** 4

**Summary:**

This paper proposes a method called Binder, which is a hierarchical concept representation method through order embedding of binary vectors. The paper explores the importance of order-based representation in natural language understanding and generation, and discusses the strengths and weaknesses of existing approaches. It also describes the geometric constraints imposed by order-based representation and how they capture semantic relationships between concepts. The paper concludes by discussing potential applications of Binder's approach to hierarchical concept representation in practical natural language processing tasks.

**Strengths:**

1. The paper proposes BINDER, a novel order embedding approach which embeds the entities at the vertex of a d-dimensional hypercube, which is simple, elegant, compact and explainable.

2. The paper proposes an optimization algorithm for BINDER, which is simple, efficient, and effective, and can be seen as a proxy of gradient descent for the combinatorial space.

3. The experimental results show that BINDER achieves great performance on link prediction and reconstruction tasks.

**Weaknesses:**

1. For reconstruction task, OE achieves better performance than BINDER with fewer dimension. Thus, BINDER does not show superiority over OE.
2. BINDER may still suffer from the limitation of optimization, leading to inferior performance.
3. It is better to report the mean results, rather than the best results.

**Questions:**

1. You claim that “In BINDER’s embedding, an ‘1’ in some representation dimension denotes “having a latent property”. How to verify it through experiments?

2. What are the differences between the proposed optimization method and randomized local search algorithm? What is the novelty of the proposed optimization method?

3. How to ensure the convergence of the proposed optimization algorithms?

4. Why BINDER is better than OE in WordNet Nouns dataset?

5. How is the dimension (in parenthesis) in Table 3 set? Why is the dimension of OE smaller than that of BINDER?

6. Can you provide the experimental results of box embedding?

---

> ### Author Response · Authors · 2023-11-19
> **Respose to Reviewer psGs**
>
> Q: For reconstruction task, OE achieves better performance than BINDER with fewer dimension. Thus, BINDER does not show superiority over OE.
>
> A: You pointed out that for reconstruction task, OE achieves better performance than BINDER for fewer dimensions and concluded that BINDER does not show superiority over OE. Your claim may be coming from the observation that on two smaller datasets (Edges: Medical: 4.3k, Music:6.5k), OE achieves 100% accuracy by using d = 10, and 20, but binder achieves 99.9% accuracy by using 50 bits.
> As a counter argument we want to clarify than BINDER uses bits, whereas other methods operates in real number domain. We need at least 4 bytes (32 bits) to represent a real number. So, the claim that OE is using fewer dimension is untrue, because OE is using d=10 (10X 32 = 320 bits), and d = 20 (20 X 32 = 640 bits), whereas BINDER is using only 50 bits. If BINDER is allowed only 50 bits, for a fair comparison other methods should be allowed $\lceil 50/32 \rceil = 2 bytes$. From our experiments we observed that even with d=5 (5x32=160 bits) and d=10 (10x32=320 bits), OE achieves 99.4% and 98.6% accuracy respectively for medical and music, which is poorer than BINDER. Most importantly, on the largest dataset (Nouns) which has 100 times more edges than these smaller datasets, BINDER achieves 99.7% accuracy with 100 bits, whereas OE achieved 96.7% accuracy by using 200 x 32 = 6400 bits.
>
> Q. BINDER may still suffer from the limitation of optimization, leading to inferior performance.
>
> A: The reconstruction performance on 5 datasets, where BINDER achieved 99.9, 99.9, 100, 100, and 99.7 percent of accuracy clearly validates that the optimization scheme that BINDER uses is excellent to find representation vector for is-a relation. We have further provided BINDER's proof of convergence in Appendix A. If you have any specific concern regarding BINDER's optimization, can you kindly elaborate so that we can alleviate your concern.
>
> Q: It is better to report the mean results, rather than the best results.
>
> A: We actually reported mean results in Table 5 along with standard deviation for all BINDER experiments. Based on Table 5 results, our best results are well within a reasonable standard deviation from the mean results.
>
> Q: Differences between the proposed optimization method and randomized local search algorithm. What is the novelty of the proposed optimization method?
>
> A: Randomized local search does not utilize gradient of the objective, whereas BINDER's optimization uses gradient of each bit position to construct a probability of flipping that bit. The idea of using gradient to flip bit probabilistically to optimize is binary space is certainly novel. The novelty of BINDER's optimization method from methodology perspective consists of: (1) compute a proxy of gradient for a binary space (Section 2.4), Compute flip probability (Section 2.5). Substantial thinking has been put forth to come up with Table 1 and Table 2 along with gradient binary logic formulas for gradient computation shown in Eq. 5-10.
>
> Q: How to ensure the convergence of the proposed optimization algorithms?
>
> A: Please see the Proof of Convergence in Appendix (Lemma 1-3 and Theorem 4)
>
> Q: Why BINDER is better than OE in WordNet Nouns dataset?
>
> A: Commenting on the performance on a specific dataset is not easy, however, we can comment based on the trend that we have observed. Nouns is the most difficult dataset due to it having 82k entities, and 743k edges; which are multiple order of magnitude larger than other datasets. Performance of all methods drops on this dataset for both the tasks. But, other methods (as well as OE) suffer more severely than BINDER. BINDER loses only one to two percent in terms of balanced accuracy, whereas other methods loses 5% to 10%, or even more on some cases.
>
> Q: How is the dimension (in parenthesis) in Table 3 set? Why is the dimension of OE smaller than that of BINDER?
>
> A: We made extensive hyper-parameter tuning for all the methods: BINDER and competitors, and reported best results for all. OE uses the smallest dimension than other competitors probably because it uses real-number space which is less constrained, whereas other competitors use hyperbolic space, which is more constrained. Comparison to BINDER, OE actually uses more dimension than BINDER, as OE uses real space and BINDER uses binary bits.
>
> Q: Provide the results of box embedding?
>
> A: Yes, we have it now: On the five datasets from Left to Right:
>
> Recon. (100, 100, 100, 100, 99.8)
> LP (0 tran:   70.4, 78.0, 74.6, 71.4, 70.8)
> LP (10 tran: 80.0, 80.0, 74.4, 74.4, 80.1)
> LP (25 tran: 80.8, 80.3, 74.6, 74.6, 86.8)
> LP (50 tran: 83.8, 84.3, 74.4, 74.5, 91.2)
>
> On LP (Link Prediction), BINDER is substantially better than Box. However, On Recon (Reconstruction), Box is marginally better than BINDER. However, LP is prediction on unknown data, but Recon is prediction on train data. So, performance on LP is more significant for a ML task.

---

> > ### Comment · Reviewer_psGs · 2023-11-23
> >
> > Thank you for your detailed response. I appreciate your efforts in addressing my concerns. I have read your response and comments of other reviewers. I suggest that the authors improve the presentation of the paper and strengthen the evaluation of the experiments.

---

### Meta-Review · Area_Chair_PLuk · 2023-12-08

**Metareview:**

The paper explores the importance of order-based representation in natural language understanding and generation, and discusses the strengths and weaknesses of existing approaches. All the reviewers find major issues about this paper and consistently agree to reject this paper. The main reasons are as follows:
1) We do not know what problem or task the proposed model is actually solving.
2) BINDER may still suffer from the limitation of optimization, leading to inferior performance.
3) Computational Efficiency: There is some operation which can be performed on the embedded representation more efficiently than on some other representation.
4) Generalization: The embedding allows one to infer missing edges between existing nodes or make predictions of graph edges from unseen nodes (based on input node features).

Please consider the detailed comments of reviewers to improve the paper.

**Justification For Why Not Higher Score:**

N/A

**Justification For Why Not Lower Score:**

N/A

---

### Decision · Program_Chairs · 2024-01-16

Reject